# Continual Learning from Simulated Interactions via Multitask Prospective Rehearsal for Bionic Limb Behavior Modeling

**Sharmita Dey**                                          *sharmita.dey@hest.ethz.ch*
*ETH Zurich*                                          *sharmita.dey@med.uni-goettingen.de*
*University of Goettingen*

**Benjamin Paassen**                                          *bpaassen@techfak.uni-bielefeld.de*
*Bielefeld University*

**Sarath Ravindran Nair**                                          *s.ravindrannair@eni-g.de*
*European Neuroscience Institute, Goettingen (ENI-G)*

**Sabri Boughorbel**                                          *sboughorbel@hbku.edu.qa*
*Qatar Computing Institute*

**Arndt F. Schilling**                                          *arndt.schilling@med.uni-goettingen.de*
*University Medical Center, Goettingen (UMG)*

**Reviewed on OpenReview:** *https://openreview.net/forum?id=Bmy82p2eez*

## Abstract

Lower limb amputations and neuromuscular impairments severely restrict mobility, necessitating advancements beyond conventional prosthetics. While motorized bionic limbs show promise, their effectiveness depends on replicating the dynamic coordination of human movement across diverse environments. In this paper, we introduce a model for human behavior in the context of bionic prosthesis control. Our approach leverages human locomotion demonstrations to learn the synergistic coupling of the lower limbs, enabling the prediction of the kinematic behavior of a missing limb during tasks such as walking, climbing inclines, and stairs. We propose a multitasking, continually adaptive model that anticipates and refines movements over time. At the core of our method is a technique called *multitask prospective rehearsal*, that anticipates and synthesizes future movements based on the previous prediction and employs a corrective mechanism for subsequent predictions. Our evolving architecture merges lightweight, task-specific modules on a shared backbone, ensuring both specificity and scalability. We validate our model through experiments on real-world human gait datasets, including transtibial amputees, across a wide range of locomotion tasks. Results demonstrate that our approach consistently outperforms baseline models, particularly in scenarios with distributional shifts, adversarial perturbations, and noise.

**Keywords:** Human Behavior Modeling, Bionics, Multitask Rehearsal, World Model, Human-machine Interaction

## 1 Introduction

Lower limb amputation and neuromuscular disorders detrimentally affect natural locomotion, compromising individuals' quality of life (Windrich et al., 2016). Those afflicted often depend on assistive technologies like prosthetics or orthoses to regain daily mobility (Windrich et al., 2016). However, conventional passive prosthetics often fall short in replicating natural gait during diverse activities, from basic transitions like standing from a seated position to more dynamic actions like running or navigating slopes (Windrich et al., 2016). The advent of powered prosthetics, equipped with integrated motors, promises a more naturalistic

gait. Nonetheless, this advancement requires a model capable of effectively approximating the complexities of human gait synergy to accurately estimate the necessary motor commands. While finite-state machines are adequate for rudimentary scenarios (Lawson, 2014; Chen et al., 2015), their construction becomes infeasible for finer gait phase resolution and multiple locomotion tasks (Lawson et al., 2013). An emerging solution lies in training models on human demonstrations to intuitively infer target limb motion (Dey et al., 2019; 2020a; 2021b). This methodology not only replicates gait's inherent fluidity but also facilitates seamless transitions across varying gaits, eliminating the need for rigid rules or heuristics. Furthermore, such learning-based gait models can be robustified by training on a large and diverse set of human demonstrations Dey et al. (2020b). To achieve personalization, these models could incorporate references that exhibit anthropomorphic features similar to those of the physically impaired subject Dey & Nair (2024).

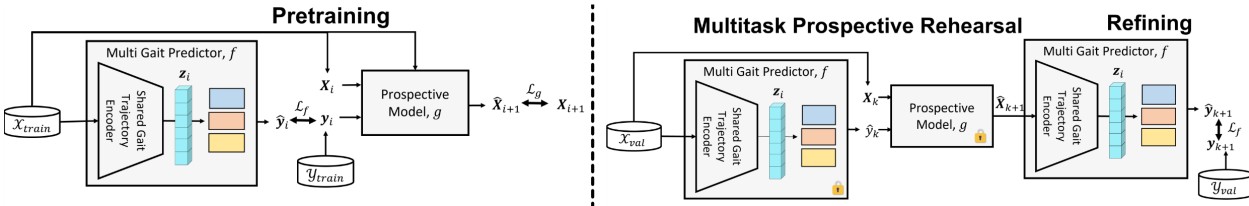

Figure 1: Overview of the multitask prospective rehearsal-based training: In the pretraining phase, the multi gait predictor model, $f$, is trained to forecast the desired joint profile $y_i$ from the history of sensor states $\{\mathbf{X}_{i-T}...\mathbf{X}_i\}$, while the prospective model $g$ aims to predict the subsequent sensor state $\mathbf{X}_{i+1}$ from $\mathbf{X}_i$ and the target joint profiles $y_i$. For prospective rehearsal, the multi gait model predictions $\hat{y}_k$ are fed along with the current sensor state $\mathbf{X}_k$ to the prospective model to project the next sensor state $\hat{\mathbf{X}}_{k+1}$. Multi gait predictor, $f$ is then refined with this projected state, $\hat{\mathbf{X}}_{k+1}, k = 1...|\mathcal{X}_{val}| - 1$ against the actual future output $y_{k+1} \in \mathcal{Y}_{val}$, to compensate the effect of its prediction errors.

Recent studies attest to the efficacy of learning-based models in gait and prosthetic behavior estimation (Dhir et al., 2018; Zabre-Gonzalez et al., 2021; Fang et al., 2020; Dey & Schilling, 2022a). However, most of these works have focused on either a single locomotion task or jointly training multiple modes in the context of multi-locomotion scenarios. Furthermore, prior research has approached multitask learning from human demonstration as a standard supervised learning task, which assumes independence across time. This oversight can cascade predictive errors, deviating significantly from the original training distribution (Kumar et al., 2022; Ross & Bagnell, 2010). Some approaches (Ross et al., 2011; Judah et al., 2014; Kelly et al., 2019) have addressed the issue of state distribution shift by using a trained model to collect additional data and label them as the model encounters new states, refining the model iteratively. However, these studies require having access to a simulator, environment, or deployment of models in the real world for data collection to estimate the expert action for the induced state. This may not be feasible or highly costly for multiple safety-critical scenarios, e.g., human-centered robotics that might entail risks of imbalance and fall. Moreover, these approaches have largely focused on single-task settings rather than continual multitask adaptation. The multifaceted dynamics of human gait, which can shift based on varying terrains, speeds, fatigue levels, or specific locomotion tasks, underscore the necessity for a model that fluidly adapts to these evolving conditions.

To achieve this objective, we present a multitask, continually adaptive gait synergy approximation model tailored for various locomotion tasks, capable of adapting to evolving gait patterns and refining itself by integrating the impact of prediction errors. Central to our method is what we call the *multitask prospective rehearsal* (Figure 1) that prepares the model to anticipate and handle potential trajectories that may arise from its predictions, creating a seamless connection between continual adaptation and the integration of prediction errors for model refinement. Unlike conventional models that often rely on retrospective analysis of movement, our method is designed to prospectively imagine and synthesize potential future locomotion patterns. By synthesizing future states, the model is, in essence, creating its own new 'unseen' data to practice on, which could lead to better generalization when encountering actual new data. The model refines its parameters incrementally as it updates itself with this enhanced data, negating the need for a simulator or new training data.

We conduct a wide range of experiments in continual and joint learning settings with different model architectures and backbones on three real-world gait datasets, including our own patient (transtibial amputee)

dataset. We establish that our model outperforms many baseline models with multiple benchmarks. Our study is the first to approach multi-gait adaptation in bionic prostheses as an error-aware multitask continual adaptation problem.

## 2    Related Work

**Prosthetic behavior models**. Conventional prosthetic behavior models primarily rely on state machines that deterministically generate control commands based on finite states within a gait cycle. These models segment the continuous gait cycle into discrete states (Lawson, 2014; Chen et al., 2015; Culver et al., 2023), which fails to accurately replicate the smooth and continuous movement patterns typical of human walking. To better model the fluidity of the gait cycle, learning-based regression methods (Dey et al., 2020b; 2021a; Dhir et al., 2018; Fang et al., 2020; Zabre-Gonzalez et al., 2021; Dey & Schilling, 2022b) have been utilized that directly predict the joint motion commands from the sensor states of residual body movement. However, these models generally adopt a straightforward reactive strategy, learning the correlation between the residual states of the body and the limb's movements. Hence, these models often oversimplify the complex nature of human motor abilities, which are inherently adaptable to various tasks, diverse environments, and evolving gait patterns.

**Multitask adaptive learning**. A principal challenge in multitask adaptive learning is the phenomenon of catastrophic forgetting (McCloskey & Cohen, 1989; Robins, 1995; French, 1999), where newly acquired knowledge can cause the loss of previously learned information. To combat this, continual learning strategies, such as regularization-based methods, architectural strategies, and experience replay (ER)(Kirkpatrick et al., 2017; Zenke et al., 2017; Smith et al., 2023) have been developed. Regularization-based methods like Elastic Weight Consolidation (EWC) (Kirkpatrick et al., 2017) and Synaptic Intelligence (SI) (Zenke et al., 2017) introduce regularization terms to guide the model towards a parameter space optimized for low error across both previous and new tasks. Architectural methods such as Progressive Neural Networks (PNNs) (Rusu et al., 2016) expand the model's structure by introducing task-specific parameters to sustain performance across tasks. Rehearsal or replay methods, including Experience Replay (ER) (Robins, 1995; Luo & Li, 2020; Li et al., 2023) and Gradient Episodic Memory (GEM) (Lopez-Paz & Ranzato, 2017), maintain a sample of previous tasks to be revisited during new task training, reinforcing the model's exposure to the old data distribution. Rehearsal approaches have empirically proved to be the most effective among several approaches developed (Merlin et al., 2022). However, these techniques have a retrospective nature and are meant for treating data samples independently. In the context of bionic prosthetics, where predictive models interact with a dynamic environment, anticipating the impact of predictions on subsequent data inputs is critical.

**Model-based reinforcement learning (RL).** Model-based RL focuses on developing a representation of the environment by gathering data from interactions driven by a specific policy. This constructed model of the environment facilitates a level of planning, enabling the estimation of potential outcomes for future states (Mnih et al., 2013; Silver et al., 2016; Racanière et al., 2017; Nagabandi et al., 2018; Chua et al., 2018). Typically, these methods rely on a reward signal and online interactions with the environment. With the advent of offline reinforcement (Zhou et al., 2021; Yu et al., 2020) and model-based imitation learning (IL) (Englert et al., 2013; Kidambi et al., 2021), access to the reward signals or online interactions can be bypassed (Hu et al., 2022). Both model-based RL and IL, by internalizing environmental dynamics, enable an agent to forecast outcomes and make informed decisions. Our approach is inspired by model-based RL; however, we use it as a rehearsal strategy within a multitask continual learning setting to mitigate forgetting, adapt to evolving tasks, and limit compounding errors, all without needing access to a reward signal or online interactions.

## 3    Method

Let $\mathbf{X}_t = \mathbf{x}_{t,1}, \ldots, \mathbf{x}_{t,N_t} \in \mathbb{R}^d$ be a time series of sensor states (3D body and joint angles, angular velocities, and linear accelerations) during a locomotion task $t$ (such as level ground walking, ascending a ramp, or climbing stairs) and let $\mathbf{y}_t = y_{t,1}, \ldots, y_{t,N_t} \in \mathbb{R}$ be the corresponding desired kinematic profiles (knee or ankle joint profiles) as demonstrated by a human. Our goal is a model $f$ which robustly predicts the correct kinematic profiles $y$ from input features $\mathbf{x}$, even under prediction errors in previous time steps, and adapts to new tasks $t$ while preserving performance on older tasks. To realize such a model, we integrate principles

from multitask adaptive learning with a prospective rehearsal approach inspired by model-based RL. Our model builds upon three core features, which we describe in turn: 1) multitask learning via a synthesis of shared learning mechanisms with task-specific adaptability, 2) continual adaptation with task rehearsal in the facet of an evolving architecture to avoid catastrophic forgetting, and 3) a prospective model, that imagines and informs about potential deviations to provide robustness against prediction errors.

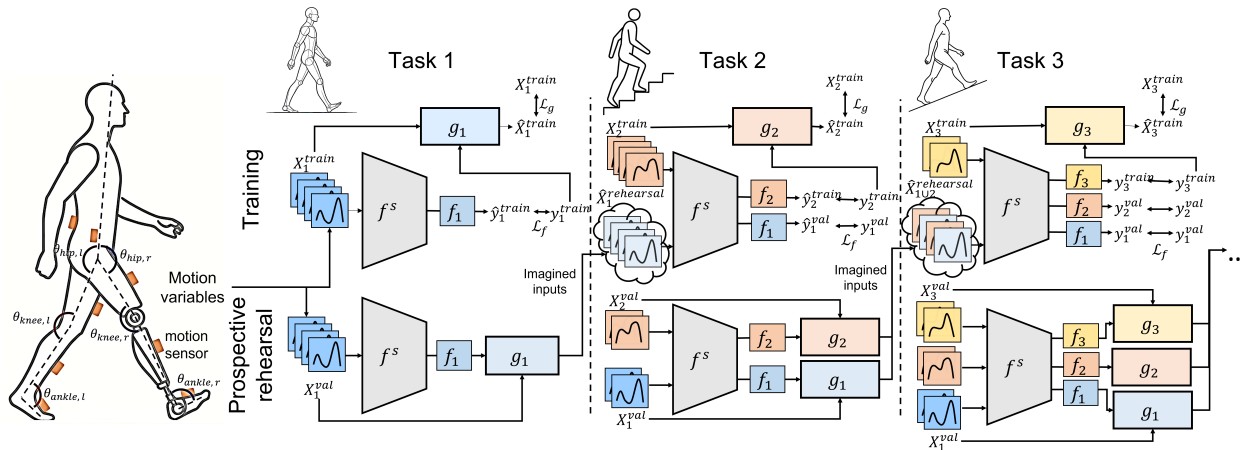

Figure 2: Summary of the overall approach. For each new task $t$, the data is split into a train $(X_t^{train}, y_t^{train})$ and validation $(X_t^{val}, y_t^{val})$ set. A task-specific layer $f_t$ is added to the shared backbone $f_s$ and trained to predict the target joint profile $y_t^{train}$. Concurrently, a prospective model for the task anticipates potential inputs based on the current joint profiles. The rehearsal data $X_t^{rehearsal}$ for all encountered tasks is generated using the prospective model imagined inputs. As new tasks are introduced, both the shared backbone $f_s$ and previously established task-specific layers are updated with this rehearsal data to facilitate continual learning.

**Multitask learning via generic and task-specific modules**. A core challenge in multitask learning is to exploit commonalities between the tasks while maintaining flexibility to account for the particularities of every single task. Our approach is to compose the gait model $f$ of two functions: a shared backbone $f^s$ across tasks, followed by task-specific layers $f_t$. The overall prediction for an input sample $\{\mathbf{x}_{t,i-T}...\mathbf{x}_{t,i}\}$ for task $t$ at time $i$ is $f(\mathbf{x}_{t,i-T}...\mathbf{x}_{t,i}) = f_t[f^s(\mathbf{x}_{t,i-T}...\mathbf{x}_{t,i})]$. This duality enables the model to distill common features across all tasks and concurrently adapt to the unique aspects of individual tasks. We employ temporal convolutions (Oord et al., 2016; Fang et al., 2020) to model the shared backbone due to their effectiveness in handling temporal sequences. To tailor the model to specific tasks, we implement task-specific layers using a two-layer feed-forward network, providing lightweight and efficient customization for each task.

**Continual learning via evolving architecture and rehearsal.** In (task) incremental adaptation, the model is trained incrementally with one task at a time. Thus, the model does not require data from all the tasks during training. When a new task $t$ is encountered, a new task-specific prediction layer $f_t$ is created and both $f_t$ and $f^s$ are trained using task-specific data. However, this method is prone to catastrophic forgetting (French, 1999; McCloskey & Cohen, 1989), that is, the model forgets the previously trained tasks when it learns a new task. To deal with the forgetting problem, we include samples from prior tasks in the training (i.e., a 'rehearsal' (van de Ven et al., 2020)). However, beyond prior rehearsal approaches, we do not merely copy training data from prior tasks. Instead, we augment it, incorporating prediction errors from the current model $f$ using the following scheme.

**Prospective model.** A principal challenge in our scenario is that prediction errors at time step $k$ may negatively impact prediction performance at time step $k + 1$, such that samples are not identically and independently distributed across time. In more detail, for the human demonstrations, we know that the sensor state $\mathbf{x}_{t,k}$ and desired kinematic profile $y_{t,k}$ is followed by the sensor state $\mathbf{x}_{t,k+1}$. However, if we predict a slightly different kinematic profile $\hat{y}_{t,k} = f(\mathbf{x}_{t,k-T}...\mathbf{x}_{t,k}) \neq y_{t,k}$, we will also observe a different sensor state $\hat{\mathbf{x}}_{t,k+1} \neq \mathbf{x}_{t,k+1}$ in the next time step, pushing us away from the training data distribution. Such deviations can accumulate over time, which is a well-known phenomenon in imitation learning (Kumar et al., 2022; Ross & Bagnell, 2010). In general, imitation learning is a helpful metaphor for our situation, as we also

wish to mimic the demonstration of human experts in a setting where independence over time does not hold. However, past theory on imitation learning has usually assumed constant bounds on the error in every time step. Such constant bounds are not realistic in our scenario with a continuous space. Unfortunately, if we slightly relax the assumption of constant bounds to Lipschitz bounds, the deviation between desired time series and predicted time series can grow exponentially, as we show in the following, simple theorem.

**Theorem 1.** *Let $x_1, \ldots, x_N \in \mathcal{X}$ be a time series from space $\mathcal{X}$, equipped with metric $d$. Further, let $f : \mathcal{X} \to \mathcal{Y}$ for some set $\mathcal{Y}$, and let $g : \mathcal{X} \times \mathcal{Y} \to \mathcal{X}$. For any $x_1' \in \mathcal{X}$, we define the time series $x_1', \ldots, x_N'$ via the recursive equation $x_{k+1}' = g[x_k', f(x_k')]$.*

*Finally, assume that for all $k \in \{1, \ldots, N-1\}$, for all $x \in \mathcal{X}$, and for some $C \in \mathbb{R}$, the following Lipschitz condition holds: $d\big(g[x, f(x)], x_{k+1}\big) \leq C \cdot d(x, x_k)$. Then, a) we obtain the exponential bound $d(x_k', x_k) \leq C^{k-1} \cdot d(x_1', x_1)$, and b) this bound is tight, i.e. there exist at least one combination of time series $x_1, \ldots, x_N$, some $f$ and $g$, as well as some $x_1'$, such that the bound holds exactly.*

*Proof.* We obtain a) via induction: The base case is $d(x_1', x_1) \leq C^0 \cdot d(x_1', x_1)$, which is trivially fulfilled. Now, assume that the claim holds for $k$. Then, for $k+1$, we obtain $d(x_{k+1}', x_{k+1}) = d\big(g[x_k', f(x_k')], x_{k+1}\big) \leq C \cdot d(x_k', x_k)$ due to the definition of $x_{k+1}'$ and the Lipschitz assumption made in the theorem. According to the induction hypothesis, the right-hand-side is bounded by $C \cdot C^{k-1} \cdot d(x_1', x_1) = C^k \cdot d(x_1', x_1)$, which concludes the induction.

b) First, note that it is sufficient to show one specific example to prove the tightness of the bound as we only claim that there exists at least one example for which the bound holds exactly. Then, assuming the setup of the example as given in the proof, we obtain the following induction. The base case is $d(x_1', x_1) \leq C^0 \cdot d(x_1', x_1)$, which is trivially fulfilled. Now, assume the claim holds for $k$. Then, for $k+1$, we obtain $d(x_{k+1}', x_{k+1}) = |x_k' + \kappa \cdot x_k' - 0| = (1 + \kappa) \cdot |x_k'| = C \cdot |x_k' - 0| = C \cdot |x_k' - x_k| = C \cdot d(x_k', x_k)$, utilizing the definitions of $x_1 = \ldots = x_N = 0$, $d(x, y) = |x - y|$, and $C = 1 + \kappa$. According to the induction hypothesis, the right-hand-side is equal to $C \cdot C^{k-1} \cdot d(x_1', x_1) = C^k \cdot d(x_1', x_1)$, which concludes the induction. Accordingly, for this example, the equality is always fulfilled exactly, implying that the bound is tight. $\square$

In this theorem, $f$ is our prediction model mapping a sensor state $\mathbf{x}_k$ to a target joint profile $y_k$, and $g$ describes the dynamics of our system, mapping the current sensor state $\mathbf{x}_k$ and joint profile $y_k$ to the next sensor state $\mathbf{x}_{k+1}$. The theorem states that, even if our predictive model $f$ is Lipschitz-bounded (i.e., it exactly matches the training data and degrades gracefully beyond the training data), the deviation can grow exponentially over time.

This observation motivates the key innovation of our work, namely a prospective rehearsal. More precisely, we train a prospective model $g$ to mimic the dynamics of the system with $g(\mathbf{x}_k, y_k) \approx \mathbf{x}_{k+1}$ and imagine the subsequent inputs resulting from the predictions of $f$. We then add data tuples $(\mathbf{x}_{k+1}', y_{k+1})$ with $\mathbf{x}_{k+1}' = g\big(\mathbf{x}_k, f(\mathbf{x}_{k-T}...\mathbf{x}_k)\big)$ to our training data. These additional training data tuples take the prediction errors of $f$ into account and thus help $f$ to counteract its own prediction errors. In our scenario, $y_{k+1}$ is the right target for the input $\mathbf{x}_{k+1}'$ because $y_{k+1}$ represents a target joint profile in a prosthetic limb. This target profile should remain the same in the same gait phase, even if $\mathbf{x}_{k+1}'$ slightly deviates from $\mathbf{x}_{k+1}$. In a general scenario, other targets might be required. Theoretically speaking, our aim is to achieve a constant bound $d\big(g[\mathbf{x}_k, f(\mathbf{x}_{k-T}, ..., \mathbf{x}_k)], \mathbf{x}_{k+1}\big) \leq \epsilon$ for small $\epsilon > 0$, such that classical imitation learning theory applies (Kumar et al., 2022; Ross & Bagnell, 2010) and an exponential deviation over time is avoided. We do so by minimizing $d\big(g[\mathbf{x}_k, y_k], \mathbf{x}_{k+1}\big)$ explicitly in Eq. equation 1.

**Summary of proposed approach.** Our overall approach is shown in Algorithm 1 and Figures 1 and 2. We add new tasks $t$ one by one and split the training data for task $t$ into a training and a validation set. We train a newly evolved task-specific layer $f_t$ and the shared backbone $f^s$ to minimize the distance between $f_t[f^s(\mathbf{x}_{t,k-T}...\mathbf{x}_{t,k})]$ and $y_{t,k}$, and, at the same time, we train the shared backbone $f^s$ and the old task layers $f_{t'}$ to minimize the distance between $f_{t'}[f^s(\mathbf{x}_{t',k-T}...\mathbf{x}_{t',k-1}, \hat{\mathbf{x}}_{t',k})]$ and $y_{t',k}$ for the data in the rehearsal buffer. The rehearsal buffer contains both the original validation samples $\mathbf{x}_{t',k}$, and the prospective samples $\hat{\mathbf{x}}_{t',k}$ for a subsample $V_{t'}$ of the time steps $k$ in the validation set. The prospective samples are generated via a task-specific, prospective model $g_{t'}$ as $\hat{\mathbf{x}}_{t',k} = g_{t'}\big(\mathbf{x}_{t',k-1}, f_{t'}[f^s(\mathbf{x}_{t',k-T-1}...\mathbf{x}_{t',k-1})]\big)$.

---

**Algorithm 1:** Multitask Prospective Rehearsal-based Adaptive Model

---

Initialize the shared backbone $f^s$.

**for** *tasks t from* $1, \ldots, L$ **do**

    Let the data for task $t$ be $(\mathbf{x}_{t,1}, y_{t,1}), \ldots, (\mathbf{x}_{t,N_t}, y_{t,N_t})$.

    Use the first $M_t$ steps as training data, the final $N_t - M_t$ steps as validation data for some $M_t < N_t$.

    Add a task-specific layer $f_t$.

    Train $f^s$ and $f_1, \ldots, f_t$ by minimizing the loss:

$$\mathcal{L}_f = \sum_{k=1}^{M_t} \| y_{t,k} - f_t[f^s(\mathbf{x}_k, ..., \mathbf{x}_{k-T})] \|^2$$

$$+ \sum_{t'=1}^{t-1} \sum_{k \in V_{t'}} \| y_{t',k} - f_{t'}[f^s(\mathbf{x}_{t',k}, ..., \mathbf{x}_{t',k-T})] \|^2 + \| y_{t',k} - f_{t'}[f^s(\hat{\mathbf{x}}_{t',k}, ..., \hat{\mathbf{x}}_{t',k-T})] \|^2$$

    Train the prospective model $g_t$ for task $t$ by minimizing the loss:

$$\mathcal{L}_g = \sum_{k=1}^{M_t - 1} \| \mathbf{x}_{t,k+1} - g_t(\mathbf{x}_{t,k}, y_{t,k}) \|^2$$

    Update the rehearsal data for all tasks.

    **for** *task t' from* $1, \ldots, t$ **do**

        $V_{t'} \leftarrow$ subsample time steps $k$ from $M_{t'}, \ldots, N_{t'} - 1$.

        **for** *time step* $k \in V_{t'}$ **do**

            $\hat{\mathbf{x}}_{t',k} = g_{t'}\big( \mathbf{x}_{t',k-1}, f_{t'}[f^s(\mathbf{x}_{t',k-T-1}, ..., \mathbf{x}_{t',k-1})] \big)$

            $V_{t'} \leftarrow (\hat{\mathbf{x}}_{t',k}, y_{t',k})$

---

## 4 Experiments

**Datasets and metrics**. We use three real-world human gait experiments' datasets. The first two, ENABL3S (Hu et al., 2018) and Embry (Embry et al., 2018), are publicly available from popular gait labs. ENABL3S comprises approximately 5.2 million training samples and 1.3 million test samples collected from ten subjects performing various locomotion activities, including level-ground walking, stair ascent and descent, and ambulating on an inclined walkway. From the Embry dataset, we selected data representing nine very distinct combinations of walking speeds (ranging from 0.8 m/s to 1.2 m/s) and inclines (ranging from -10 degrees to +10 degrees). This selection resulted in approximately 512,000 training samples and 104,000 test samples from ten subjects, encompassing a diverse range of movements. The third dataset is a novel collection from our lab, focusing on gait patterns of patients (transtibial amputees), recorded with a 200 Hz infrared motion capture system using twelve cameras, and includes activities like walking, stair ascent, and descent, with around 27K training and 11K testing samples. Ethical clearance for these experiments was obtained from our institutional review board. This dataset, along with the others, includes motion-related variables such as 3D angles of body joints and segments, velocities, and accelerations. To evaluate the accuracy of the desired joint profiles, $y_k$, we calculate the coefficient of determination ($R^2$), between the ground truth trajectories and model-based joint profiles.

**Ablations**. We assess the contribution of each component on overall performance. We explore both joint and continual learning paradigms, comparing models with shared final layers against those with task-specific final layers Dey et al. (2024). Further, we consider both *conventional* and *prospective rehearsal* with each of the shared and task-specific final layer variants. In joint training, the training data is either augmented with the multitask prospective rehearsal from a validation set or simply with the original samples from the validation set itself (conventional rehearsal). In all conditions, we used the same amount of training data to maintain fairness. Augmentation with prospective rehearsal improved the performance of models with both shared and task-specific final layers. Further, the results provide evidence that task-specific final layers help

Table 1: Experimental results for (A) joint multitask learning using different prediction head architectures and replays, (B) task incremental learning using different prediction head architectures and replays, (C) different continual learning strategies – SI (Synaptic Intelligence), EWC (Elastic weight consolidation), PNN (Progressive Neural Networks), GEM (Gradient episodic memory), ER (Experience Replay), and proposed prospective replay, (D) data augmentation with Gaussian noise vs. prospective replay buffer. (LW=Level walking, RA=Ramp ascent, RD=Ramp descent, SA=Stair ascent, SD=Stair descent, Spd=Speed in m/s, Inc=Inclination in °)

| Dataset | | ENABL3S | | | | | Embry et. al. 2018 | | | | | | | | | Transtibial amputees | | |
|---|---|---|---|---|---|---|---|---|---|---|---|---|---|---|---|---|---|---|
| Task | | LW | RA | RD | SA | SD | Spd 0.8 / Inc -10 | 0.8 / 0 | 0.8 / 10 | 1.0 / -10 | 1.0 / 0 | 1.0 / 10 | 1.2 / -10 | 1.2 / 0 | 1.2 / 10 | LW | SA | SD |
| head | replay | **(A) Joint multi-task learning** | | | | | | | | | | | | | | | | |
| S | conventional | 0.83 | 0.87 | 0.89 | 0.91 | 0.86 | 0.86 | 0.88 | 0.91 | 0.91 | 0.94 | 0.94 | 0.85 | 0.93 | 0.93 | 0.79 | 0.79 | 0.89 |
| S | prospective (ours) | 0.89 | 0.91 | 0.92 | 0.93 | 0.88 | 0.88 | 0.92 | 0.92 | 0.93 | 0.96 | 0.94 | 0.89 | 0.96 | 0.94 | 0.93 | 0.72 | 0.90 |
| T | conventional | 0.90 | **0.94** | 0.95 | **0.96** | 0.93 | 0.95 | **0.98** | 0.94 | 0.95 | **0.99** | 0.96 | 0.80 | 0.96 | **0.98** | 0.97 | 0.82 | 0.92 |
| T | prospective (ours) | **0.92** | **0.94** | **0.96** | **0.96** | **0.94** | **0.96** | **0.98** | **0.96** | **0.96** | **0.99** | **0.97** | **0.90** | **0.97** | **0.98** | **0.98** | **0.91** | **0.94** |
| head | replay | **(B) Task-incremental learning** | | | | | | | | | | | | | | | | |
| S | conventional | 0.80 | 0.86 | 0.84 | 0.89 | 0.83 | 0.76 | 0.87 | 0.83 | 0.81 | 0.90 | 0.90 | 0.74 | 0.90 | 0.88 | 0.94 | 0.83 | 0.68 |
| S | prospective (ours) | 0.84 | 0.90 | 0.89 | 0.91 | 0.89 | 0.90 | 0.92 | 0.92 | 0.94 | 0.96 | 0.95 | 0.89 | **0.97** | 0.95 | 0.94 | 0.85 | 0.72 |
| T | conventional | 0.88 | 0.93 | 0.93 | 0.94 | 0.92 | 0.92 | **0.98** | 0.93 | 0.94 | 0.98 | **0.97** | 0.87 | **0.97** | 0.97 | 0.97 | **0.91** | 0.91 |
| T | prospective (ours) | **0.91** | **0.95** | **0.95** | **0.97** | **0.94** | **0.96** | **0.98** | **0.95** | **0.97** | **0.99** | **0.97** | **0.91** | **0.97** | **0.98** | **0.98** | **0.91** | **0.92** |
| **(C) Continual learning strategies** | | | | | | | | | | | | | | | | | | |
| SI | | 0.06 | 0.45 | 0.51 | 0.55 | 0.93 | -0.38 | 0.33 | 0.59 | 0.26 | 0.40 | 0.83 | 0.34 | 0.66 | 0.97 | 0.37 | 0.31 | 0.92 |
| EWC | | 0.13 | 0.31 | 0.55 | 0.52 | 0.92 | 0.04 | 0.33 | 0.65 | 0.08 | 0.54 | 0.92 | 0.53 | 0.77 | 0.96 | -0.30 | -0.57 | 0.85 |
| PNN | | 0.90 | 0.91 | 0.94 | 0.93 | 0.91 | 0.85 | 0.97 | **0.96** | 0.88 | 0.97 | 0.96 | 0.87 | 0.96 | **0.98** | 0.68 | 0.82 | 0.79 |
| GEM | | 0.76 | 0.91 | 0.92 | 0.95 | 0.93 | 0.88 | 0.97 | 0.93 | 0.90 | 0.97 | 0.96 | 0.84 | **0.97** | 0.97 | 0.82 | 0.80 | 0.86 |
| ER | | 0.88 | 0.93 | 0.93 | 0.94 | 0.92 | 0.92 | **0.98** | 0.93 | 0.94 | 0.98 | **0.97** | 0.87 | **0.97** | 0.97 | 0.97 | **0.91** | 0.91 |
| prospective (ours) | | **0.91** | **0.95** | **0.95** | **0.97** | **0.94** | **0.96** | **0.98** | 0.95 | **0.97** | **0.99** | **0.97** | **0.91** | **0.97** | **0.98** | **0.98** | **0.91** | **0.92** |
| **(D) Data augmentation** | | | | | | | | | | | | | | | | | | |
| Gaussian noise inj. | | 0.88 | 0.93 | 0.93 | 0.94 | 0.91 | 0.90 | 0.97 | 0.94 | 0.91 | 0.97 | 0.96 | 0.81 | **0.97** | 0.97 | 0.92 | 0.83 | 0.88 |
| prospective (ours) | | **0.91** | **0.95** | **0.95** | **0.97** | **0.94** | **0.96** | **0.98** | **0.95** | **0.97** | **0.99** | **0.97** | **0.91** | **0.97** | **0.98** | **0.98** | **0.91** | **0.92** |

to account for particularities among tasks. A comparison of performance in task-incremental learning yields similar results. The results demonstrate that the choice of a combination of prospective rehearsal and an evolving architecture, where task-specific final layers are integrated with a shared backbone, yields the best performance across the board (Table 1A–B). All following experiments are performed with task-specific final layers.

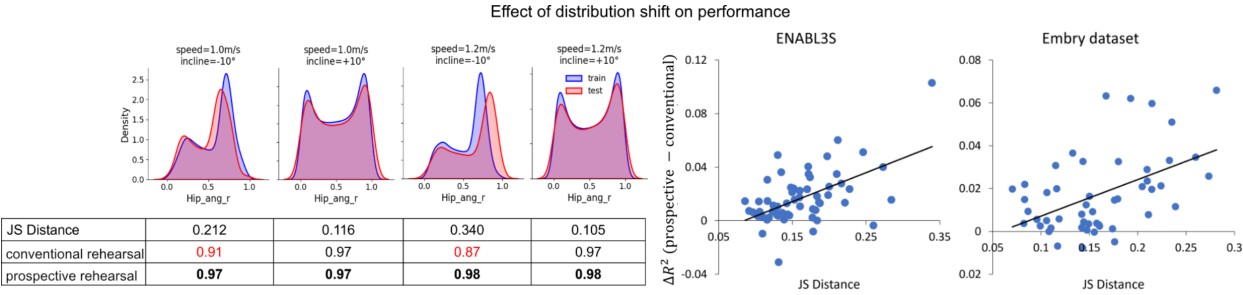

Figure 3: Comparison of prospective rehearsal with a conventional rehearsal in the face of distribution shift. (Left) Training/test data distributions, their Jenson-Shannon (JS) distances, and $R^2$ predictions on four different tasks from the Embry dataset show the prospective rehearsal's efficacy. (Right) A graph plots the $R^2$ difference between methods against JS distance, with a linear trendline indicating that the prospective rehearsal's benefits increase with greater distribution divergence.

**Benchmarking continual learning strategies**. In the continual learning paradigm, we compare our approach against regularization strategies like SI and EWC, architectural strategies, PNNs, and state-of-the-art replay-based strategies like GEM and the classic form of experience replay, which we refer to as conventional rehearsal (Table 1C). We observe that prospective rehearsal performs best across the board, indicating better task memory retention and generalization. In Table 1D, we compare data augmentation via Gaussian noise injection to the proposed prospective rehearsal, with prospective rehearsal performing best across the board. All results are statistically significant (see appendix).

**Resilience to distribution shift**. In some cases, we do not observe a significant improvement in the performance of the prospective rehearsal over a conventional rehearsal. To explain these results, we inspect the difference in $R^2$ between the prospective and conventional rehearsal versus the Jenson-Shannon (JS) distance (Endres & Schindelin, 2003) between training and test distributions (Figure 3). We observe that the error difference positively correlates with JS distance. In other words, the advantage of the prospective rehearsal becomes more pronounced the more training and test distribution deviate. This is in line with our theoretical analysis: The more training and test distribution differ, the more prediction errors of $f$ we expect, which means that our proposed prospective rehearsal helps more. Since real-world prosthesis data is prone to noise and variability from various exteroceptive and interoceptive factors (Prahm et al., 2019), being robust to distribution shifts is an important requirement for this domain, and our approach performs well in this regard.

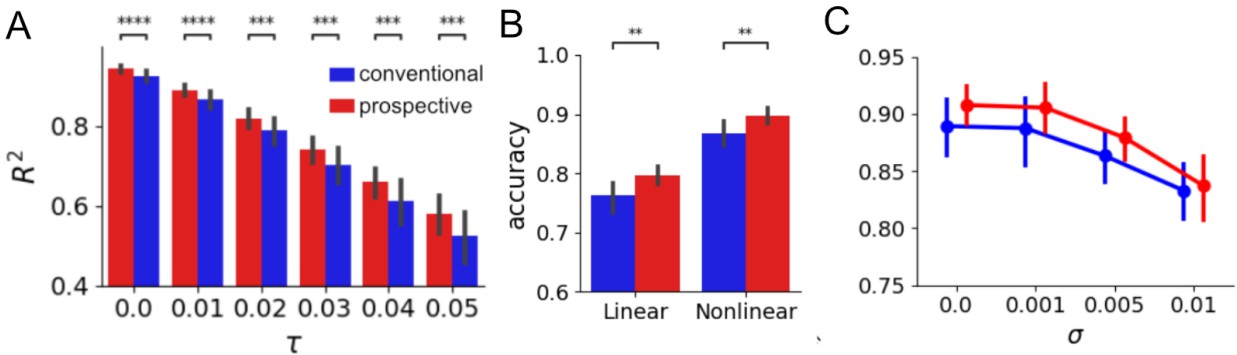

Figure 4: A) Performance comparison of prospective and conventional rehearsal strategies across different strengths, $\tau$, of adversarial perturbations. B) Performance of linear and non-linear probing on downstream locomotion task classification. Prospective rehearsal training performs best across the board. (C) Comparison of the performance of continual models with conventional and prospective rehearsal for different levels of input noise. Results are presented for models with task-specific final layers.

**Resilience to adversarial perturbations**. To further assess the resilience of the prospective rehearsal approach, we compare its outcomes to those without it in the face of adversarial perturbations. These adversarial samples are created using the Fast Gradient Sign Method (FGSM) (Goodfellow et al., 2014) across diverse epsilon thresholds, $\tau$. Notably, our prospective rehearsal technique surpasses the conventional baseline across all evaluated FGSM attack intensities (Figure 4A). Furthermore, as the perturbation intensity escalates, the performance disparity between the prospective and the conventional method widens, reinforcing the superior adaptability of prospective rehearsal under scenarios where test data distribution deviates from the training set. The results are statistically significant.

**Probing downstream performance**. We evaluate the representation capacity of models trained with prospective rehearsal techniques versus those trained without it, focusing on a downstream challenge—classifying different types of locomotion. We employ both linear and nonlinear (MLP) probes to assess the representation of the shared layers, denoted as $f_s(\mathbf{x}_{t,k-T}...\mathbf{x}_{t,k})$, from models with task-specific heads aimed at classifying the task $t$. The findings indicate that prospective rehearsal yields significantly better task classification performance than its conventional counterpart for both linear and nonlinear probes (Figure 4B). This result underscores that the prospective model-based training confers an added benefit of robustness and adaptability compared to conventional rehearsal.

**Resilience to noise**. Further, we analyze the robustness of our proposed model against perturbations to the input signals during test time in the form of multivariate white Gaussian noise, $\mathcal{N}(\mathbf{0}, \mathbf{\Sigma} \in \mathbb{R}^{d \times d})$. We apply different levels of perturbation $L$, where $L$ is defined as the ratio of the standard deviation of the Gaussian noise to the standard deviation of the test inputs. Figure 4C shows the $R^2$ of our proposed rehearsal and a conventional rehearsal for five different tasks on the ENABL3S dataset. While performance generally decreases with higher amounts of noise, the prospective rehearsal yields better $R^2$ across the board.

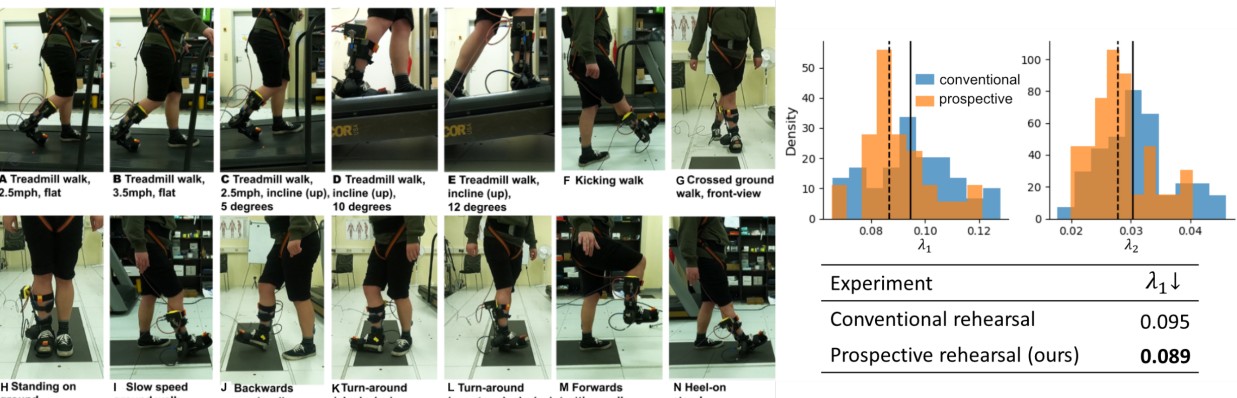

Figure 5: (Left) Snapshots from the experiments on a subject. Different motion conditions such as walking at different speeds and inclines were tested on an instrumented treadmill with adjustable inclines (A–E). Furthermore, ground walking on a level walkway with different motion maneuvers was also tested (F–N). (Right) Stability comparison of foot kinematic profiles yielded using prospective and conventional rehearsals. First $\lambda_1$ and second $\lambda_2$ Lyapunov exponents (lower is better), computed using the Eckmann method, indicate that the foot kinematic profiles obtained using our proposed method exhibit greater stability compared to those obtained without employing our method. Median Lyapunov exponent of the prospective rehearsal method, (dotted line) is lower compared to conventional rehearsal (solid line).

**Gait stability assessment with an exoskeleton**. In a pilot study, able-bodied individuals walked with an ankle exoskeleton (Figure 5 left) controlled by joint profiles predicted by the model with the prospective rehearsal approach. Participants reported that the predictions felt timely, accurate, and in sync with their residual body movements. To evaluate the stability of the model-based foot profiles, with and without the prospective rehearsal approach, we measured the maximal Lyapunov exponent using the Eckmann method (Eckmann et al., 1986). The Lyapunov exponent measures the rate of divergence of nearby trajectories in a dynamical system, with lower values indicating higher stability. Our results showed that both the first and second Lyapunov exponents were lower when using our proposed method, indicating more stable gait trajectories (Figure 5 right).

## 5 Conclusion

Our research introduces a novel framework for bionic prostheses' application, adept at handling multiple locomotion tasks, adapting progressively, foreseeing movements, and refining. Central to our method, is a novel prospective rehearsal training scheme that, through empirical studies on diverse datasets, reliably outperforms standard techniques, particularly under challenging conditions of adversaries, distribution shifts, noise, and task transfer. Notably, the prospective rehearsal approach is data-centric, hence, model-agnostic, and can be applicable across various model architectures. While these results are promising, a limitation of our study is the absence of comprehensive clinical trials, largely due to the rigorous and extensive ethical approval process required for such research. Recognizing this, our future work is set to expand upon these findings through more extensive trials, including clinical trials and in-home studies, to validate and refine our model in everyday settings. This adaptable, robust system signals a significant leap forward for prosthetic gait prediction in dynamic settings, offering improved quality of life for individuals with lower limb impairments.

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

# A Additional Results

## A.1 Model architecture

In Table 2, we show the results of using different architectures for the shared backbone $f^s$. All models were designed to have approximately the same number of parameters. Since TCN tends to perform best across datasets, we opt for TCN as the backbone architecture in practice.

| Dataset | ENABL3S | | | | | | Embry et. al. 2018 | | | | | | | | | Transtibial amputees | | |
|---|---|---|---|---|---|---|---|---|---|---|---|---|---|---|---|---|---|---|
| Task | LW | RA | RD | SA | SD | Spd | 0.8 | 0.8 | 0.8 | 1.0 | 1.0 | 1.0 | 1.2 | 1.2 | 1.2 | LW | SA | SD |
| | | | | | | Inc | -10 | 0 | 10 | -10 | 0 | 10 | -10 | 0 | 10 | | | |
| Linear | 0.68 | 0.89 | 0.89 | 0.90 | 0.83 | | 0.04 | 0.15 | 0.51 | 0.04 | 0.12 | 0.55 | 0.13 | 0.11 | 0.55 | 0.39 | 0.64 | 0.68 |
| MLP | 0.67 | 0.91 | 0.91 | 0.91 | 0.86 | | -0.06 | 0.14 | 0.52 | 0.06 | 0.13 | 0.55 | 0.11 | 0.12 | 0.56 | 0.42 | 0.56 | 0.70 |
| LSTM | 0.88 | 0.94 | 0.94 | 0.96 | 0.93 | | 0.86 | 0.97 | 0.95 | 0.83 | 0.96 | 0.96 | 0.82 | 0.95 | 0.97 | 0.41 | 0.59 | 0.64 |
| Transformer | 0.9 | 0.94 | **0.95** | 0.96 | 0.93 | | 0.88 | 0.97 | **0.96** | 0.86 | 0.98 | **0.97** | 0.88 | **0.97** | **0.98** | **0.98** | 0.84 | 0.92 |
| TCN | **0.91** | **0.95** | 0.95 | **0.97** | **0.94** | | **0.96** | **0.98** | 0.95 | **0.97** | **0.99** | 0.97 | **0.91** | 0.97 | 0.98 | **0.98** | **0.91** | **0.92** |

Table 2: Comparison of various shared module architectures on different locomotion conditions from three datasets.

## A.2 Model predictions

In Figure 6 (left), We show the model-based motion trajectories of a transtibial amputee subject's hypothetical ankle, as demonstrated by our multitask model using prospective rehearsal. The alignment between the predicted and biological able-bodied limb motions suggests the model's effectiveness in approximating gait synergy across various lower limb joints and segments, aiming to replicate normal locomotion for impaired patients. Figure 6 (right) presents analogous results for an able-bodied subject, using their own limb motion as the target. The close match between model-based and natural limb motions further underscores the model's potential in generating movement signals for bionic limbs, offering hope for enhancing mobility in individuals with lower-extremity disabilities.

In Figure 7 we show the example predictions from baseline models as compared to our approach. The joint angular positions predicted by the baseline models (shared final layer model and EWC regularization) showed frequent misalignments with sharp disagreements between the predicted and actual trajectories. In a safety-critical scenario such as prosthesis control for locomotion assistance, such misalignments can be very costly as they can lead to imbalance. On the other hand, predictions from our proposed model show a higher degree of alignment with the ground truth (actual) trajectories, which is necessary to ensure smoother locomotion using a prosthetic device controlled by these predictions.

## A.3 Resilience to distribution shift

We show the effectiveness of our prospective rehearsal-based approach in dealing with distribution shifts during test time from the lens of model predictions. Figure 8 illustrates the predictions of models trained with conventional rehearsal and our prospective rehearsal for two scenarios—one with a low and the other with a high distribution shift during test time. As anticipated, both models exhibit comparable performance when the test and training distributions are similar (see the bottom of Figure 8). However, in the presence of a distribution shift at test time, the model trained with conventional rehearsal shows a decline in performance, while the prospective rehearsal-trained model sustains its predictive accuracy (see the top of Figure 8).

## A.4 Stability against forgetting

We explore the ability to preserve the knowledge of previously learned tasks after learning new ones. Figure 9A shows how the NRMSE values (lower is better) for each task develop when training subsequent tasks, both for a shared final layer (top) and for a task-specific final layer (bottom). Both architectures preserve the memory of the previous tasks, but the performance on old tasks slightly worsens for a shared final layer, whereas it remains the same for task-specific final layers.

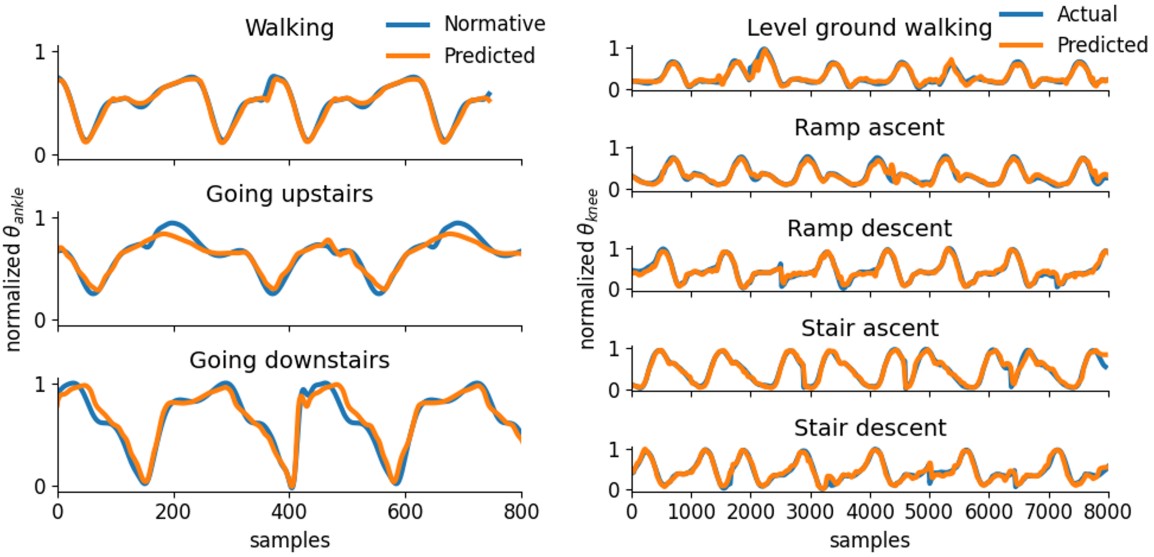

Figure 6: Model-based kinematic profiles from our proposed rehearsal-based models. (Left) Motion trajectories of a transtibial amputee subject's hypothetical ankle, predicted by our model. The normative (target) trajectories are computed from an able-bodied individual with anthropometric features and walking speed similar to that of the amputee subject. (Right) Predicted motion trajectories of an able-bodied subject from ENABL3S dataset for different locomotion modes. The target trajectories in this case are the subject's own limb motion. In both cases, the predicted trajectories are highly aligned with the target limb motion.

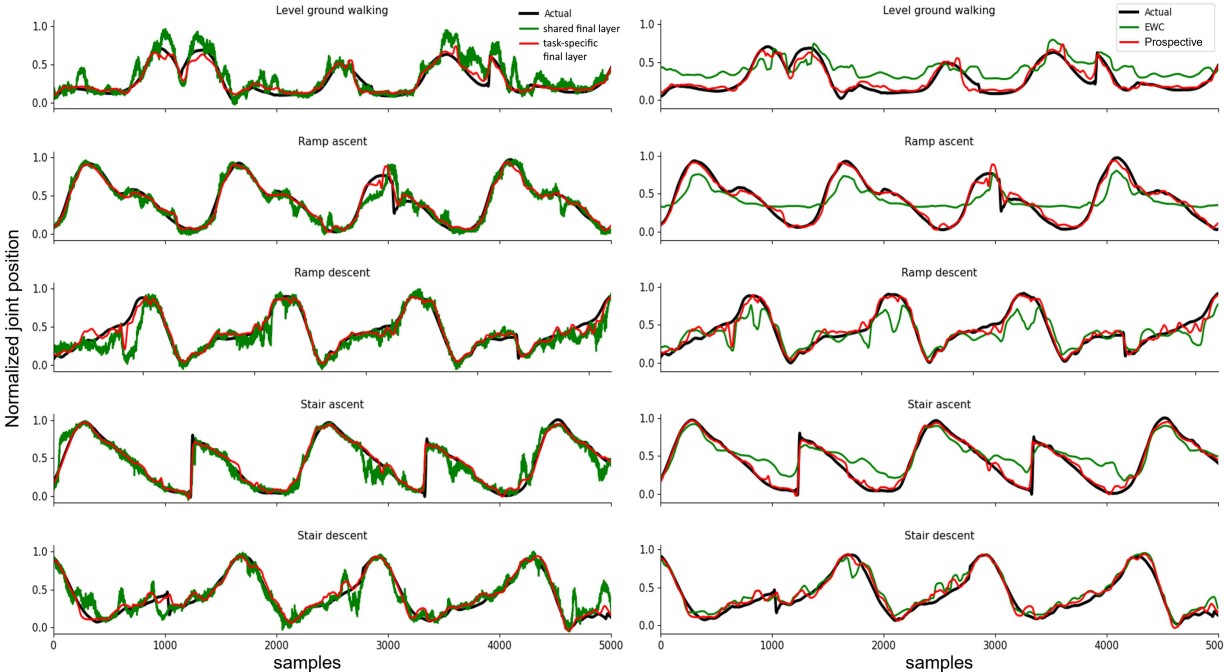

Figure 7: Baseline prediction examples. (Left) Examples of joint position trajectories for different locomotion tasks predicted by a shared final layer model vs. task-specific final layer model. (Right) Examples of joint position trajectories predicted by a task-specific final layer model with EWC regularization vs. task-specific final layer model with prospective rehearsal. The black curve shows the ground truth trajectories.

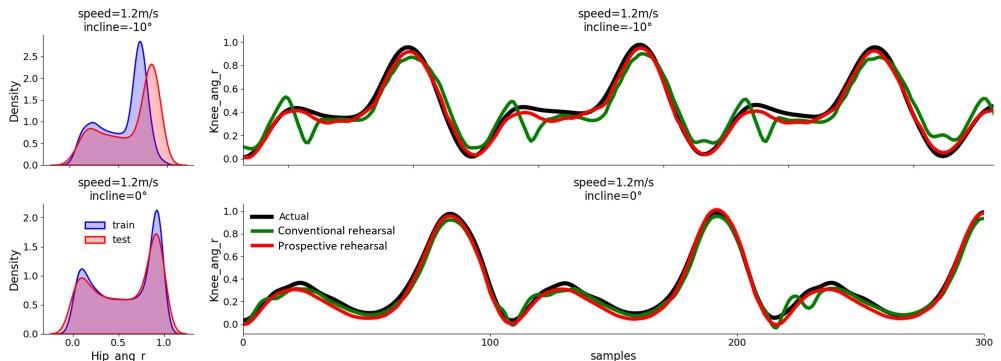

Figure 8: (Left) Train and test distributions for two tasks from Embry et. al. dataset for a representative subject. (Right) Predictions of model trained with conventional and prospective rehearsal.

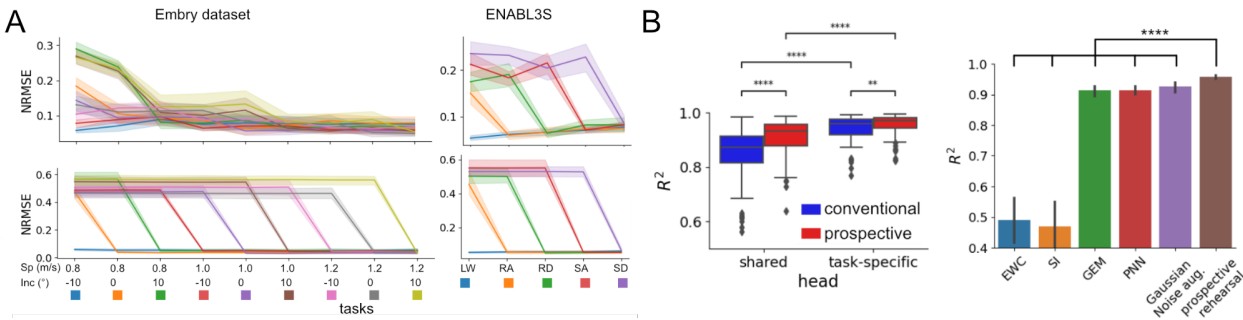

Figure 9: (A) Plot depicting NRMSE values for test tasks from Embry and ENABL3S datasets post-training, with the X-axis for learned tasks, color for test tasks, and shading indicating standard error across subjects. (B) (Left) Results of statistical significance tests comparing the performance of conventional and proposed prospective rehearsal for shared and task-specific final layers. (Right) Comparison of performance of the proposed prospective rehearsal with state-of-the-art continual learning strategies. Statistical significance was assessed using the Wilcoxon-signed rank test with Bonferroni correction, **** : $p < 1e - 4$,*** : $p < 1e - 3$,** : $p < 1e - 2$.

|  |  | Train task | | | | |
|---|---|---|---|---|---|---|
|  |  | Walk | Ramp up | Ramp down | Stair up | Stair down |
|  | Walk | **0.054** | 0.14 | 0.16 | 0.177 | 0.18 |
|  | Ramp up | 0.454 | **0.055** | 0.147 | 0.134 | 0.176 |
| Test task | Ramp down | 0.517 | 0.517 | **0.055** | 0.165 | 0.156 |
|  | Stair up | 0.516 | 0.516 | 0.516 | **0.054** | 0.193 |
|  | Stair down | 0.506 | 0.506 | 0.502 | 0.505 | **0.066** |

Table 3: RMSE values of a model with task-specific prediction heads trained without replay on different tasks after it is trained for a particular task.

We further show through ablation studies that, both the rehearsal and architectural choice are necessary for preventing the forgetting problem (see Tables 3–5). A model trained without replay (Tab. 3) excels only on its current task, but suffers from catastrophic forgetting on previous tasks. In contrast, models with replay (Tab. 4 and 5) reduce forgetting. Furthermore, using task-specific prediction heads outperforms a shared head, further mitigating forgetting. Thus, both replay and task-specific heads help address forgetting.

In addition, to systematically analyze the model performance as it is trained for multiple tasks, we also computed two metrics (similar to Díaz-Rodríguez et al. (2018)) to measure how well the model performs on previously trained tasks after training it with new tasks.

|  |  | Train task | | | | |
|  |  | Walk | Ramp up | Ramp down | Stair up | Stair down |
|---|---|---|---|---|---|---|
|  | Walk | **0.054** | **0.062** | **0.067** | **0.071** | **0.078** |
|  | Ramp up | 0.15 | **0.06** | **0.067** | **0.074** | **0.075** |
| Test task | Ramp down | 0.175 | 0.19 | **0.064** | **0.083** | **0.083** |
|  | Stair up | 0.213 | 0.184 | 0.215 | **0.071** | **0.083** |
|  | Stair down | 0.235 | 0.231 | 0.204 | 0.228 | **0.086** |

Table 4: RMSE values of a model with task-shared prediction heads trained with replay on different tasks after it is trained for a particular task.

|  |  | Train task | | | | |
|  |  | Walk | Ramp up | Ramp down | Stair up | Stair down |
|---|---|---|---|---|---|---|
|  | Walk | **0.053** | **0.057** | **0.058** | **0.059** | **0.062** |
|  | Ramp up | 0.454 | **0.057** | **0.055** | **0.056** | **0.056** |
| Test task | Ramp down | 0.502 | 0.501 | **0.051** | **0.056** | **0.057** |
|  | Stair up | 0.553 | 0.551 | 0.551 | **0.053** | **0.054** |
|  | Stair down | 0.53 | 0.53 | 0.53 | 0.53 | **0.066** |

Table 5: RMSE values of a model with task-specific prediction heads trained with replay on different tasks after it is trained for a particular task.

The backward transfer is computed as

$$\text{BWT}(t) = \frac{1}{t-1} \sum_{j=1}^{t-1} \varepsilon_j(j) - \varepsilon_t(j), \tag{1}$$

where $t$ is the latest task the model is trained for, $\varepsilon_i(j)$ is the prediction error on task $j$ once the model was trained with the task $i$. $\text{BWT}(t)$ computes the change in performance of the model on previously trained tasks, $j = 1..t-1$ after it is trained for a new task $t = 2..T$. Larger values of backward transfer are preferred.

The forgetting ratio for a task $t$ is computed as

$$\text{FR}(t) = \frac{\varepsilon_T(t) - \varepsilon_t(t)}{\varepsilon_t(t)}, \tag{2}$$

where $T$ represents the final task on which the model was trained. $\text{FR}(t)$ gives the relative change in performance of the model on task, $t = 1..T-1$, after it was trained with all available tasks. Smaller values of forgetting ratio are preferred.

| Train task, $t$ | Walk | Ramp up | Ramp down | Stair up | Stair down |
|---|---|---|---|---|---|
| no replay | – | -0.043 | -0.065 | -0.079 | -0.098 |
| task-shared with replay | – | -0.004 | -0.006 | -0.013 | -0.014 |
| **task-specific with replay** | – | **-0.002** | **-0.001** | **-0.002** | **-0.003** |

Table 6: Backward knowledge transfer, $\text{BWT}(t)$, (transfer of knowledge to previously learned tasks) when the model is trained with a new task, $t$. Higher the better.

| Evaluation task, $t$ | Walk | Ramp up | Ramp down | Stair up | Stair down |
|---|---|---|---|---|---|
| no replay | 2.33 | 2.29 | 1.99 | 2.64 | – |
| task-shared with replay | 0.42 | 0.25 | 0.27 | 0.16 | – |
| **task-specific with replay** | **0.16** | **0** | **0.1** | **-0.02** | – |

Table 7: Forgetting ratios, $\text{FR}(t)$, of the model on a task, t after training it with all available tasks. The lower the better.

Tables 6 and 7 show the backward transfer and forgetting ratios of models trained without replay, and the ones trained with replay with different architectures. These results further illustrate that the models with

task-specific prediction heads and replay perform best in the face of continual task learning and dealing with catastrophic forgetting.

### A.5   Statistical significance of results

A Wilcoxon-signed rank test revealed that the proposed rehearsal significantly outperforms conventional rehearsal, both for shared as well as task-specific final layers (Figure 9B). Further, task-specific final layers significantly outperform a shared final layer, irrespective of rehearsal strategy. This shows that both the evolving architecture as well as the prospective rehearsal strongly contribute to the improvement in performance using the proposed method. Our experiments also show that the proposed rehearsal significantly outperforms many state-of-the-art continual learning strategies and data augmentation with Gaussian noise.

## B   Real-world Deployment of Models: Exoskeleton/Hardware Experiments

### B.1   Experimental setup

The experimental setup for the real-world deployment of the proposed models is depicted in Figure 10. Utilizing Inertial Measurement Units (IMUs) to monitor the user's residual body motion, our model could infer and predict the ankle-joint positions in real-time. These predictions were then employed as control inputs for the ankle exoskeleton. The ankle exoskeleton, configured in position control mode, received foot angular position predictions generated by the gait-predictive model to regulate its movements. The gait-predictive model, trained and stored on a computer, produced joint position predictions corresponding to the movement of the residual body. These predictions were transmitted through a CAN communication system to actuate the ankle exoskeleton motor.

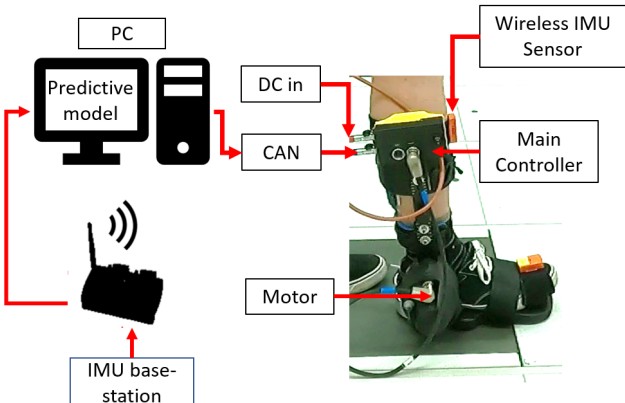

Figure 10: Communication pipeline of the ankle exoskeleton is shown. The ankle exoskeleton is equipped with a DC power inlet and a CAN communication inlet to interact with the main controller of the ankle exoskeleton. First, the IMU sensor signals are received by the computer (PC) via the IMU base station. The IMU signals are processed in real-time, and the trained model stored on the PC generates predictions to control the ankle exoskeleton. The predictions are sent to actuate the ankle exoskeleton via the CAN communication protocol to the main controller of the ankle exoskeleton, which in turn sends commands to actuate the device's motor.

Besides the quantitative stability comparison of gait as reported in the main paper, we also analyzed data on trunk posture during two experimental scenarios: one with the subjects walking using the model-predicted ankle positions with the exoskeleton, and another where they walked without any assistance. We additionally looked at the variability in trunk kinematics, which is another critical indicator of balanced gait (Rábago et al., 2015). Figure 11 shows the lateral trunk angles in the case where a subject walked freely (blue) and when the subject walked with the assisted powered exoskeleton, whose ankle-joint positions were predicted from

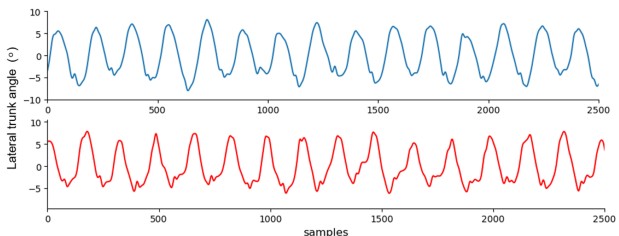

Figure 11: Lateral trunk angles when a subject walked freely (top) and when the subject walked with the assisted powered exoskeleton, whose ankle-joint positions were predicted from our model (bottom).

our model (red). Our findings revealed that the lateral trunk angles during assisted and unassisted walking were comparable, suggesting that the exoskeleton, guided by our model's predictions, did not necessitate any compensatory movements from the subjects. This similarity implies that our model can effectively predict control signals that facilitate a stable and comfortable walking experience.

## C   Baselines

### C.1   Continual learning strategies

We compared our method to other continual learning strategies such as synaptic intelligence (SI) (Zenke et al., 2017), elastic weight consolidation (EWC) (Kirkpatrick et al., 2017), experience replay (ER), and gradient episodic memory (GEM)(Lopez-Paz & Ranzato, 2017).

**SI** is a regularization-based continual learning strategy aimed at overcoming catastrophic forgetting, which is the tendency of neural networks to completely forget previously learned information upon learning new data. SI achieves this by measuring the importance of synaptic parameters to past tasks and selectively constraining the update of these important parameters when new tasks are learned. This method estimates a "synaptic importance" score for each parameter of the neural network, based on how much a change in that parameter would affect the total loss across all tasks learned so far. When new data is encountered, SI allows the network to update primarily those parameters that are less important for the tasks learned previously, thereby preserving the performance on those tasks.

**EWC** is another regularization-based continual learning strategy that addresses the problem of catastrophic forgetting in neural networks. EWC works by calculating the importance of each parameter to the old tasks and then applying a regularization term that penalizes changes to the most crucial parameters. The 'importance' is quantified using the Fisher Information Matrix, which captures how sensitive the model's predictions are to changes in each parameter. When the model is trained on a new task, parameters important for past tasks are kept relatively stable, while less important parameters are more free to change. This method is inspired by principles of neuroplasticity in the human brain, where certain synaptic pathways are consolidated and become less plastic after learning, preserving essential knowledge.

**ER** combats forgetting by intermittently retraining the model on a random subset of previously seen examples. This approach mimics the way humans recall and reinforce knowledge over time by revisiting past experiences. In practice, ER maintains a memory buffer that stores a collection of past data samples. During the training of new tasks, the learning algorithm interleaves the new data with samples drawn from this memory buffer. By doing so, the model is regularly reminded of the old tasks while it learns the new ones, which helps to maintain its performance across all tasks. The process of experience replay is a simple yet effective way to preserve old knowledge in the neural network without compromising the learning of new information.

**GEM** is a strategy designed to mitigate forgetting by leveraging experiences stored in a memory. GEM maintains a subset of the data from previously encountered tasks in an episodic memory and uses this stored data to guide the learning process for new tasks. When a new task is introduced, GEM compares the gradients of the loss with respect to the current task and the gradients concerning the tasks stored in

memory. It then adjusts the update rule to ensure that the loss of the episodic memory does not increase, effectively preventing the network from unlearning the previous tasks. This is achieved through a constrained optimization problem that allows the model to learn the new task while not getting worse on the tasks stored in memory.

**PNN** is an architectural approach to overcoming the challenges of catastrophic forgetting in machine learning, particularly when dealing with sequential or multitask learning scenarios. By architecturally encapsulating knowledge gained from previous tasks, PNNs enable the integration of new information without overwriting what has been previously learned. This is achieved through the use of separate neural network columns for each task, where each new column is connected to all previous ones via lateral connections. These connections allow the networks to access previously learned features, making PNNs highly effective for tasks requiring the retention and transfer of knowledge across different domains.

### C.2 Adversarial perturbations

The Fast Gradient Sign Method (FGSM) attack used for crafting our adversarial perturbations is a method for generating adversarial examples based on the gradients of the neural network. It perturbs an original input by adding noise that is determined by the sign of the input's gradient with respect to the neural network's parameters. This method is designed to be quick and effective, typically requiring only one step to create an adversarial example.

To elaborate, the FGSM takes the original input $x$, the target label $y$, and the model parameters $\theta$, and then it computes the gradient of the loss with respect to $x$. The noise is then generated by taking the sign of this gradient and multiplying it by a small factor, $\tau$, which controls the intensity of the perturbation. The adversarial example is created by simply adding this perturbation to the original input:

$$x' = x + \tau \cdot sign(\nabla_x J(\theta, x, y))$$

where $\nabla_x J(\theta, x, y)$ is the gradient of the model's loss with respect to the input, $\theta$ represents the model's parameters, and $\tau$ is a small constant. FGSM can be used to reveal the vulnerabilities of a neural network and is a common benchmark for evaluating the robustness of machine learning models against adversarial attacks.

## D  Training and Hyperparameters

| Training Hyperparameters | Value |
| --- | --- |
| Optimizer | SGD |
| Learning rate | 1e-4 |
| Momentum | 0.9 |
| Early stopping patience | 10 |
| Batch size | 100 |
| Training steps | 130k (ENABL3) / 13k (Embry) / 1.3k (Amputees) |
| Max. rehearsal buffer size | 3k |
| Task balancing | yes |

Table 8: Hyperparameters used to train our model

**Rehearsal size** To ensure a balanced comparison, we maintained an identical rehearsal size for the conventional rehearsal, prospective rehearsal, and Gaussian noise augmentation approaches. This measure was taken to avoid attributing any performance improvement to data volume inflation, which could result from augmentation. By sampling an equivalent quantity of data for both conventional and prospective rehearsal, we aimed to neutralize the potential impact of having more or less data on performance outcomes.

**Rehearsal sampling strategy** We employ a task-balanced reservoir sampling strategy for rehearsal, which ensures that samples from each task are maintained in a balanced manner over time. Specifically, this approach allows us to continuously update the rehearsal buffer with representative samples from all tasks,

without favoring recent tasks over older ones. During each minibatch update, a balanced subset of samples from all tasks is used to ensure that the model's performance does not degrade on earlier tasks. This method effectively mitigates the issue of bias towards more recent tasks. Moreover, our strategy aligns with well-established practices in continual learning, where task-balanced replay buffers are commonly used to prevent catastrophic forgetting Merlin et al. (2022); Krawczyk & Gepperth (2024).

The hyperparameters employed for training the model are detailed in Table 8.

## E   Broader Impact

The strategies introduced in this paper have broad implications across human-centered assistive technologies, intelligent interfaces, and autonomous systems. Beyond prosthetics, this framework can improve the traversability of autonomous agents, such as legged robots Dey et al. (2022); Morrell et al. (2024) and self-driving vehicles Schmid et al. (2022), by enabling them to predict and adapt to dynamic environments. Furthermore, the ability to anticipate and account for the prediction errors is valuable for rehabilitation robotics and personalized healthcare solutions, offering more responsive and robust systems for patient care. Additionally, these modeling techniques may be adapted for industrial robotics to enhance safety in human-robot collaboration and for teleoperation scenarios that demand precise movement coordination. Taken together, this research represents a step toward resilient, human-centric AI technologies with wide-reaching societal benefits.

