# OpenReview forum: "Continual Learning from Simulated Interactions via Multitask Prospective Rehearsal for Bionic Limb Behavior Modeling"
_TMLR — Accepted by TMLR_

### Review · Reviewer_t3mt · 2024-10-13

**Summary Of Contributions:**

The paper studies the human behavior modeling in bionic prosthesis control, focusing on a multitask, continually adaptive framework.   The authors propose a "multitask prospective rehearsal" approach to handle multitask learning and continual adaptation while mitigating distribution shifts between training and testing samples. The method relies on a rehearsal strategy similar to experience replay or generative experience replay in continual learning. The model features both a shared backbone and task-specific modules for different tasks. A key contribution is the rehearsal buffer, which contains both real and prospective samples generated by a task-specific model. Experimental results demonstrate that the proposed method outperforms previous baselines, particularly in dealing with distribution shifts, noise, and adversarial attacks.

**Audience:**

Yes

**Broader Impact Concerns:**

It would be beneficial for the authors to include a Broader Impact Statement discussing the ethical implications of applying machine learning models in medical prosthetics, with particular focus on patient safety and data privacy.

**Claims And Evidence:**

Yes

**Requested Changes:**

1.	Provide a more general proof or rewrite Theory 1b to better reflect the specific scenario discussed.
2.	Explain why the prediction problem does not use a roll-out strategy for future m-step predictions to reduce error, as is common in time-series prediction tasks.  It is better to try a roll-out strategy or provide reasons for not using it.
3.	Discuss related works on generative replay and selective sampling strategies to provide a more comprehensive background.

Please refer to the details in the "Weaknesses" section for specifics.

**Strengths And Weaknesses:**

[Strengths]
1. The paper tackles an important and practical problem in human behavior modeling for bionic prosthesis control, which has implications for machine learning applications in healthcare and real-world prediction problems.
2. The proposed method, multitask prospective rehearsal, introduces a reasonable approach for continual learning, enhancing experience replay with prospective data generation.
3. The experimental results clearly demonstrate the effectiveness of the approach, showing strong performance across multiple benchmarks.

[Weaknesses]
1. The proof of Theory 1b is only provided for a very specific case ($x_1 = x_2 = … = x_N = 0,  g(x,y) = x+y$). This does not constitute a strict proof. The authors should either generalize the proof or reframe Theory 1b to be more specific and avoid overly general terms.
2. It is unclear why the prediction problem in the paper was not trained using a roll-out strategy or multi-step prediction, similar to model-predictive control or standard time-series prediction methods. It is better to try a roll-out strategy or provide reasons for not using it.  For example, in Algorithm 1, the loss functions Lf and Lg do not incorporate a roll-out mechanism for multi-step prediction, which is commonly used to reduce prediction errors.  As an example  (where $\hat{x}$ and $\hat{y}$ is predicted value. )
$$
L_g^{rollout} = \sum_{k=1}^{M_t-1} \left\| x_{t, k+1} - g_t \left( x_{t, k}, y_{t, k} \right) \right\|^2 + \sum_{i=1}^{m} \sum_{k=1}^{M_t-1} \left\| {x_{t, k+i+1}}   - g_t \left( \hat{x_{t, k+i}}, \hat{y_{t, k+i}} \right)   \right\|^2 $$

3. Some relevant works are missing from the discussion. The replay strategy with augmentation in this paper is more akin to generative replay (see [1, 2]) than traditional experience replay. Additionally, selective sampling strategies (see [3]) have been proposed to enhance performance in rehearsal-based methods. These works should be discussed.

[1] Shin, Hanul, et al. "Continual learning with deep generative replay." Advances in neural information processing systems, 2017.
[2] Van de Ven, et al. "Brain-inspired replay for continual learning with artificial neural networks." Nature communications, 2020.
[3] Yin, Peng, et al. "Bioslam: A bioinspired lifelong memory system for general place recognition." IEEE Transactions on Robotics, 2023  .

---

> ### Author Response · Authors · 2024-11-03
> **Rebuttal**
>
> Thank you sincerely for your helpful review and valuable suggestions on our manuscript. Below, we respond to each of your comments in detail.
>
> **1. Theory 1b.** We thank the reviewer for considering the details of our proof. There might be a misunderstanding here: the b) part of Theorem 1 claims "this bound is tight, i.e. there exists some time series $x_1, ..., x_N$, some $f$ and $g$, as well as some $x_1'$, such that the bound holds exactly." We provide exactly one such combination, thus fulfilling the proof. However, to avoid any misunderstandings, in line with your suggestion, we will adapt the phrasing of the theorem to: "This bound is tight, i.e. there exists at least one combination of time series $x_1, ..., x_N$, f, g, and $x_1'$, such that the bound holds exactly."
>
> **2. M-step rollouts.** Thank you for this valuable suggestion; we agree that a multi-step roll-out strategy is useful. However, the primary goal of our paper was to demonstrate that even a simple, single-step approach leveraging model-generated synthetic dynamics could effectively enhance robustness and performance under potential perturbations. We aimed to showcase the impact of this straightforward strategy, leaving a full multi-step rollout as future work.
>
> In line with your feedback, we conducted additional experiments with multi-step rollouts, applying our proposed method autoregressively across multiple steps. This approach yielded performance gains, especially on timesteps further into the future. Notably, our model trained with single-step prospective rehearsal maintained its performance well even when rolled out multiple steps into the future, while the model trained with a 10-step rollout consistently performed robustly across extended rollouts.
>
> | **timestep**             | **1** | **2** | **3** | **4** | **5** | **6** | **7** | **8** | **9** | **10** |
> |--------------------------|-------|-------|-------|-------|-------|-------|-------|-------|-------|--------|
> | **prospective (1-step)** | 0.92  | 0.92  | 0.92  | 0.92  | 0.91  | 0.9   | 0.89  | 0.87  | 0.86  | 0.85   |
> | **10-step rollout**      | 0.93  | 0.93  | 0.93  | 0.93  | 0.93  | 0.92  | 0.91  | 0.91  | 0.9   | 0.9    |
>
> **3. Additional literature.** We thank the reviewer for highlighting additional relevant literature in the area of replay strategies. We will incorporate these suggestions and discuss them in the revised paper to provide a more comprehensive background.
>
> *Generative replay relevance:* We agree that our use of prospective rehearsal has parallels with generative replay, especially in terms of generating new rehearsal samples that account for model predictions rather than just sampling past experiences. Inspired by generative replay, our method does not simply rely on stored samples but employs a predictive model to project possible future states based on current predictions, allowing the model to continuously adapt.
>
> *Selective sampling strategies:* We acknowledge the utility of selective sampling in rehearsal-based continual learning and will incorporate a discussion of it. Although our approach does not directly employ selective sampling, it aligns with the concept by replaying samples that account for prediction errors, effectively sampling “future critical states” that the model anticipates needing for continual adaptation.
>
> Following the reviewer’s suggestion, we will reference works like Shin et al. (2017), Van de Ven et al. (2020), and Peng, et al. (2023) in our related work.

---

### Review · Reviewer_Eak7 · 2024-10-20

**Summary Of Contributions:**

The authors present a multitask, continually adaptive gait synergy approximation model that adapts to various locomotion tasks and evolving gait patterns by integrating prediction errors into its learning process. The key insight of the proposed work is the multitask prospective rehearsal technique, which allows the model to anticipate and adjust for potential future trajectories, enhancing the connection between continual adaptation and error integration. The model generates synthetic future states, generating 'unseen' data from previous task model to improve its generalization capability. The proposed work is evaluated on extensive experiments across continual and joint learning settings on three real-world gait datasets, including a unique transtibial amputee dataset. This study is the first to address multi-gait adaptation in bionic prostheses using an error-aware multitask continual adaptation framework.

**Audience:**

Yes

**Broader Impact Concerns:**

I am concerned about the practical value of the proposed framework, as it does not seem to address key challenges like imbalance or fall risks. Moreover, its performance on public datasets is not competitive compared to existing methods.

**Claims And Evidence:**

Yes

**Requested Changes:**

1.Conduct Ablation Studies to Address Concerns on Catastrophic Forgetting (Weakness 1)

2. Design Ablation Studies to Address Concerns on the Bias in Rehearsal Data Augmentation (Weakness 2)

3. Provide a Robust Solution and Ablation Studies for Error Accumulation in Temporal Dynamics (Weakness 3)

4. Clarify and Manage Assumptions of Lipschitz Bounds (Weakness 4)

5. Address Scalability Issues with Task-Specific Layers (Weakness 5)

6. Clarify the Strategy for Task Sampling in Rehearsal (Weakness 6)

7. Include Comparisons with Existing Work (Weakness 7)

**Strengths And Weaknesses:**

**Strengths**:

1. The authors introduce a novel multitask prospective rehearsal pipeline, allowing it to anticipate and adjust for future trajectories, improving adaptation over time for locomotion predictions.

2. The proposed method synthesizes future states, effectively generating new ‘unseen’ data for training, which helps enhance generalization capabilities when faced with new conditions.

3. This is the first study to tackle multi-gait adaptation in bionic prostheses using an error-aware multitask continual adaptation framework, offering new insights in the field.

**Weaknesses**:

1. **Catastrophic Forgetting in Continual Learning**: the proposed method to task-specific adaptation involves developing a new task-specific prediction layer for each new task while training both the shared backbone and task-specific layers using only the new task data. This is inherently prone to catastrophic forgetting, where learning new tasks causes the model to forget previously learned tasks. However, even with the rehearsal approach (including data from previous tasks), the model may still lose performance on earlier tasks due to insufficient rehearsal samples or overfitting to new tasks. The evolving architecture could also make it difficult to balance learning between new and old tasks, especially as the number of tasks increases. I suggest that the authors conduct ablation studies to analyze the performance drop on each task, which will help to better understand this issue.

2. **Rehearsal Approach and Data Augmentation**: the method augments rehearsal data using prediction errors generated by the current model. However, this method may introduce biased training samples that might not accurately reflect the true task distribution. By relying on augmented samples based on the model’s own predictions, the model could end up reinforcing its existing errors. This could lead to a divergence between the model’s predictions and actual human demonstrations, especially if the initial errors are significant or poorly handled. The ablation studies should be designed to address concerns about the accumulation of errors from the previous model's augmentation.

3. **Impact of Prediction Errors on Temporal Dynamics**: the approach recognizes that prediction errors at one time step can affect subsequent time steps, causing deviations from the true data distribution. This is a fundamental problem in time-series prediction and imitation learning, where small errors can compound over time. While this issue is well-documented, the method does not provide a robust solution for mitigating the accumulation of errors over time. It mentions relaxing error bounds to Lipschitz bounds, but this can lead to exponential growth in deviations between the predicted and desired time series. Without addressing how to control or correct these deviations effectively, the method may struggle with long-term stability. Meanwhile, Theorem 1 is not clearly defined, and its proof lacks rigor. A proper theoretical proof should include a thorough mathematical derivation that clearly connects the assumptions to the conclusion.

4. **Assumptions of Lipschitz Bounds in Prediction Errors**: the discussion suggests that constant error bounds are unrealistic in continuous spaces and instead considers Lipschitz bounds for the errors. However, this may lead to exponential divergence between predicted and actual outcomes over time. The assumption of Lipschitz continuity implies that small changes in input can lead to proportional changes in output, but in a time-series context, this means that errors can amplify exponentially. Without a clear strategy for managing this, the model’s predictions may become increasingly inaccurate, limiting its applicability to longer sequences.

5. **Potential Scalability Issues with Task-Specific Layers**: the proposed framework developed new task-specific layers for each task, which may lead to scalability issues as the number of tasks grows. If the number of tasks increases significantly, the model size and complexity could grow, leading to higher computational and memory demands. This could make the model less practical for real-world applications where new tasks are frequently introduced. Thus, I suggest conducting an ablation study that demonstrates the performance of each task over time, rather than just focusing on the overall performance across the entire dataset.

6. **Unclear Strategy for Task Sampling in Rehearsal**: the authors mentions including samples from previous tasks during training for rehearsal but does not clarify how the task samples are selected or balanced in the training process. The effectiveness of rehearsal in preventing catastrophic forgetting depends on how representative the rehearsal samples are of prior tasks. Without a clear sampling strategy, the rehearsal process might be biased towards more recent tasks, reducing its ability to preserve performance on earlier tasks.

7. **No Comparison with The Existing Work**: there is no comparison with relevant existing work, such as "Ensemble diverse hypotheses and knowledge distillation for unsupervised cross-subject adaptation" (Information Fusion, 2023). Given that the authors conducted all experiments on widely used open-source datasets like ENABL3S, it is important to include comparisons with other studies that address the same topic.

---

> ### Author Response · Authors · 2024-11-03
> **Rebuttal: part 1**
>
> We appreciate the reviewer’s time and effort in reviewing our manuscript and providing insightful feedback and suggestions for improvement. Below, we address each of your comments point-by-point.
>
> **1. Catastrophic forgetting in continual learning.** Thank you for your thoughtful feedback. The rationale behind our architecture is to balance task-specific adaptability with knowledge sharing. Our model employs a shared core to capture generalities across diverse motion conditions by leveraging shared weights, while, to meet the specific needs of individual tasks, we incorporate lightweight task-specific layers on top of the shared core. This approach mirrors the design seen in models like Dinov2 and has been shown to be effective, as also illustrated in our ablation studies. In Table 1 of our manuscript, we show that in both joint and task-incremental learning settings, models with task-specific adaptation outperform models with only shared layers across all the datasets (Enabl3S, Embry et al. and Transtibial Amputees).
>
> Regarding rehearsal, it is widely acknowledged as one of the most practical strategies for continual learning to mitigate catastrophic forgetting. To maintain balanced rehearsal samples across tasks, we employ a task-balanced rehearsal buffer [1], [2], following common practices in the field. Specifically, during each minibatch update, a balanced replay buffer is used to ensure fair task updates.
>
> In line with your suggestion, we are happy to include tabular results of how performance drops without using the aforementioned strategies, as shown below.
>
> - *RMSE* values of a model with task-specific prediction heads trained *without replay* on different tasks after it is trained for a particular task (Tab. 1).
>
> |               |                |           |             |   Train task  |              |                |
> |---------------|----------------|:---------:|:-----------:|:-------------:|:------------:|:--------------:|
> |               |                |  **Walk** | **Ramp up** | **Ramp down** | **Stair up** | **Stair down** |
> |               	| **Walk**       | **0.054** |     0.14    |      0.16     |     0.177    |      0.18      |
> |               	| **Ramp up**    |   0.454   |  **0.055**  |     0.147     |     0.134    |      0.176     |
> | **Test task** 	| **Ramp down**  |   0.517   |    0.517    |   **0.055**   |     0.165    |      0.156     |
> |               	| **Stair up**   	|   0.516   |    0.516    |     0.516     |   **0.054**  |      0.193     |
> |               	| **Stair down** |   0.506   |    0.506    |     0.502     |     0.505    |    **0.066**   |
>
> - *RMSE* values of a model with *task-shared* prediction heads trained *with replay* on different tasks after it is trained for a particular task (Tab. 2).
>
> |               |                |           |             |   Train task  |              |                |
> |---------------|----------------|:---------:|:-----------:|:-------------:|:------------:|:--------------:|
> |               |                |  **Walk** | **Ramp up** | **Ramp down** | **Stair up** | **Stair down** |
> |               | **Walk**       | **0.054** |  **0.062**  |   **0.067**   |   **0.071**  |    **0.078**   |
> |               | **Ramp up**    |    0.15   |   **0.06**  |   **0.067**   |   **0.074**  |    **0.075**   |
> | **Test task** | **Ramp down**  |   0.175   |     0.19    |   **0.064**   |   **0.083**  |    **0.083**   |
> |               | **Stair up**   |   0.213   |    0.184    |     0.215     |   **0.071**  |    **0.083**   |
> |               | **Stair down** |   0.235   |    0.231    |     0.204     |     0.228    |    **0.086**   |
>
> - *RMSE* values of a model with *task-specific* prediction heads trained *with replay* on different tasks after it is trained for a particular task (Tab. 3).
>
> |  |  |  | |   Train task  | | |
> |---------------|----------------|:---------:|:-----------:|:-------------:|:------------:|:--------------:|
> |               |                | **Walk**  | **Ramp up** | **Ramp down** | **Stair up** | **Stair down** |
> |               | **Walk**       | **0.053** | **0.057**   | **0.058**     | **0.059**    | **0.062**      |
> |               | **Ramp up**    | 0.454     | **0.057**   | **0.055**     | **0.056**    | **0.056**      |
> | **Test task** | **Ramp down**  | 0.502     | 0.501       | **0.051**     | **0.056**    | **0.057**      |
> |               | **Stair up**   | 0.553     | 0.551       | 0.551         | **0.053**    | **0.054**      |
> |               | **Stair down** | 0.53      | 0.53        | 0.53          | 0.53         | **0.066**      |
>
>
> A model trained without replay (Tab. 1) excels only on its current task, but suffers from catastrophic forgetting on previous tasks. In contrast, models with replay (Tab. 2 and 3) reduce forgetting. Furthermore, using task-specific prediction heads outperforms a shared head, further mitigating forgetting. Thus, both replay and task-specific heads help address forgetting.

---

> ### Author Response · Authors · 2024-11-03
> **Rebuttal: part 2**
>
> **Continuation from rebuttal part 1**
>
> In addition, to systematically analyze the model performance as it is trained for multiple tasks, we also computed two metrics (similar to [3]) to measure how well the model performs on previously trained tasks after training it with new tasks.
>
> 1) The backward transfer is computed as
> $$
>     \mathit{BWT(t)} = \frac{1}{t-1}\sum_{j=1}^{t-1}\varepsilon_j(j) - \varepsilon_t(j)
> $$
> where $t$ is the latest task the model is trained for,
> $\varepsilon_i(j)$ is the prediction error on task $j$ once the model was trained with the task $i$. $\mathit{BWT(t)}$ computes the change in performance of the model on previously trained tasks $j=\{1..t-1\}$ after it is trained for a new task $t =\{2...T\}$. Larger values of backward transfer are preferred.
>
> 2) The forgetting ratio for a task $t$ is computed as
> $$
>     \mathit{FR(t)} = \frac{\varepsilon_T(t)-\varepsilon_t(t)}{\varepsilon_t(t)}
> $$
>
> where $T$ represents the final task on which the model was trained. $FR(t)$ gives the relative change in performance of the model on task, $t = \{1...T-1\}$, after it was trained with all available tasks. Smaller values of forgetting ratio are preferred.
>
> Following tables show the backward transfer and forgetting ratios of models trained without replay, and the ones trained with replay with different architectures.
>
>
> - Backward knowledge transfer, $BWT(t)$, (transfer of knowledge to previously learned tasks) when the model is trained with a new task, t (Tab. 4)
>
> | **Train task, t**             | **Walk**    | **Ramp up**     | **Ramp down**   | **Stair up**    | **Stair down**  |
> |-------------------------------|-------------|-----------------|-----------------|-----------------|-----------------|
> | **no replay**                 | --          | -0.043          | -0.065          | -0.079          | -0.098          |
> | **task-shared with replay**   | --          | -0.004          | -0.006          | -0.013          | -0.014          |
> | **task-specific with replay** | -- | **-0.002** | **-0.001** | **-0.002** | **-0.003** |
>
> - Forgetting ratios, $FR(t)$, of the model on a task, t after training it with all available tasks (Tab. 5)
>
> | **Evaluation task, t**        | **Walk**      | **Ramp up** | **Ramp down** | **Stair up**   | **Stair down** |
> |-------------------------------|---------------|-------------|---------------|----------------|------------------------------------|
> | **no replay**                 | 2.33          | 2.29        | 1.99          | 2.64           | --                                 |
> | **task-shared with replay**   | 0.42          | 0.25        | 0.27          | 0.16           | --                                 |
> | **task-specific with replay** | **0.16** | **0**  | **0.1**  | **-0.02** | --                                 |
>
> As can be seen, models with task-specific prediction heads and replay perform best in the face of continual task learning and dealing with catastrophic forgetting.
>
> **2. Rehearsal approach and data augmentation:** We thank the reviewer for pointing us to this potential misunderstanding: while the predicted next time steps may be beyond the distribution of the original training data, they do not risk compounding error effects because the targets/labels in our training data stem from the original training data distribution. In other words: The augmented samples are precisely designed to guide the system back toward the task distribution as represented by the training samples. This may also help to resolve some of the confusion regarding our theoretical argument. We are happy to make this point more explicit.

---

> ### Author Response · Authors · 2024-11-03
> **Rebuttal: part 3**
>
> **3 and 4. Error accumulation, assumptions of Lipschitz bounds, mitigation:** We agree with the reviewer that the problem is exactly the exponential growth behavior due to compound errors over time. This is what Theorem 1 shows: Exponential growth due to compound errors. We are happy to adapt the text to make this more clear.
>
> We are also happy to expand the proof by providing the full inductions for the directions of the proof, a) and b). These inductions work as follows:
>
> a) The base case is $d(x_1', x_1) \leq C^0 \cdot d(x_1',
> x_1)$, which is trivially fulfilled. Now, assume that the claim holds for $k$. Then, for $k+1$, we
> obtain $d(x_{k+1}', x_{k+1}) = d(g[x_k', f(x_k')], x_{k+1}) \leq C \cdot
> d(x_k', x_k)$ due to the definition of $x_{k+1}'$ and the Lipschitz assumption made in the theorem. According to the  induction hypothesis, the right-hand-side is bounded by $C \cdot C^{k-1} \cdot d(x_1', x_1) = C^k \cdot d(x_1', x_1)$, which concludes the induction.
>
> b) First, note that it is sufficient to show one specific example to prove the tightness of the bound as we only claim that there exists at least one example for which the bound holds exactly. Then, assuming the setup of the example as given in the proof, we obtain the following
> induction. The base case is $d(x_1', x_1) = C^0 \cdot d(x_1', x_1)$, which is trivially fulfilled. Now, assume the claim holds for $k$. Then, for $k+1$, we obtain $d(x_{k+1}', x_{k+1}) = |x_k' + \kappa \cdot x_k' - 0| = (1+\kappa) \cdot |x_k'| = C \cdot |x_k' - 0| = C \cdot |x_k' - x_k| = C \cdot
> d(x_k', x_k)$, utilizing the definitions of $x_1 = \ldots = x_N = 0$, $d(x, y) = |x - y|$, and $C = (1 + \kappa)$. According to the induction hypothesis, the right-hand-side is equal to $C \cdot C^{k-1} \cdot d(x_1', x_1) = C^k \cdot d(x_1', x_1)$, which concludes the induction. Accordingly, for this example, the equality is always fulfilled exactly, implying that the bound is tight.
>
> *Regarding the mitigation strategy:* This is achieved by virtue of anticipating prediction errors, augmenting the training data accordingly. Thus, we hope to achieve a constant error bound instead of a Lipschitz bound, which makes classic imitation learning theory applicable (which guarantees polynomial bounds, not exponential ones). We are happy to adapt the text to clarify this argument.
>
> **5. Potential scalability issues with task-specific layers:**  As mentioned earlier, the task-specific layers in our approach are designed to balance knowledge sharing and task-specific adaptation, a strategy also employed in models like Dinov2. Our experiments demonstrate that these task-specific layers improve performance. To manage the increase in tasks, task-specific layers can be merged based on task similarity [4], which can help limit the linear growth of the model. For example, a task-specific layer can be added only when the similarity between tasks exceeds a certain threshold.  We will clarify this in the discussion and propose it as an avenue for future work.
>
> **7. Comparison with [5]:** We appreciate your recommendation to include comparisons with [5]. However, it is important to note that the suggested work focuses on locomotive intent classification, which is a discrete task. In contrast, our method directly predicts and regresses locomotion variables, meaning it operates in a continuous output space. This requires the model to generate real-valued predictions over time, such as joint angles, which, generally, involve a higher level of complexity. Due to this fundamental difference in task complexity and formulation, a direct comparison is not feasible.
>
> That said, we have already compared our method with several well-established continual learning approaches, as shown in **Table 1, row 3** of our paper. We benchmarked our approach against five existing continual learning strategies, including replay-based, regularization-based, and architectural methods (such as Synaptic Intelligence (SI), Elastic Weight Consolidation (EWC), Progressive Neural Networks (PNN), Gradient Episodic Memory (GEM), and Experience Replay (ER)). Our results demonstrate that our proposed prospective rehearsal augmentation outperforms these methods across various metrics.
>
> We will ensure to make this more explicit in the final version of the paper and highlight the significance of our method in relation to existing continual learning strategies.

---

> ### Author Response · Authors · 2024-11-03
> **Rebuttal: part 4**
>
> **6. Strategy for Task Sampling in Rehearsal:** We employ a task-balanced reservoir sampling strategy for rehearsal, which ensures that samples from each task are maintained in a balanced manner over time. Specifically, this approach allows us to continuously update the rehearsal buffer with representative samples from all tasks, without favoring recent tasks over older ones. During each minibatch update, a balanced subset of samples from all tasks is used to ensure that the model's performance does not degrade on earlier tasks. This method effectively mitigates the issue of bias towards more recent tasks. Moreover, our strategy aligns with well-established practices in continual learning, where task-balanced replay buffers are commonly used to prevent catastrophic forgetting [1], [2]
>
> We will further clarify this sampling strategy in the paper and provide more detailed discussions on how task balance is achieved during the rehearsal process to ensure the preservation of performance across all tasks.
>
>
> References:
>
>
> [1] Merlin, Gabriele, et al. "Practical recommendations for replay-based continual learning methods." International Conference on Image Analysis and Processing. Cham: Springer International Publishing, 2022.
>
> [2] Krawczyk, A., & Gepperth, A. (2024). An Analysis of Best-practice Strategies for Replay and Rehearsal in Continual Learning. In Proceedings of the IEEE/CVF Conference on Computer Vision and Pattern Recognition (pp. 4196-4204).
>
> [3] Díaz-Rodríguez, Natalia, et al. "Don't forget, there is more than forgetting: new metrics for Continual Learning." arXiv preprint arXiv:1810.13166 (2018).
>
> [4] Dwivedi, Kshitij, and Gemma Roig. "Representation similarity analysis for efficient task taxonomy & transfer learning." In Proceedings of the IEEE/CVF Conference on Computer Vision and Pattern Recognition, pp. 12387-12396. 2019.
>
> [5] Zhang, Kuangen, et al. "Ensemble diverse hypotheses and knowledge distillation for unsupervised cross-subject adaptation." Information Fusion 93 (2023): 268-281.

---

### Review · Reviewer_EQtz · 2024-10-28

**Summary Of Contributions:**

This paper presents a multitask, continually adaptive model for human gait prediction, targeting the control of bionic prosthetics, particularly for transtibial amputees. The model utilizes “multitask prospective rehearsal,” a novel approach that anticipates future movements, corrects errors incrementally, and adapts across varied gait patterns and locomotion tasks. Experiments demonstrate the model’s superior performance over baseline models, especially under conditions of distributional shifts and adversarial perturbations.

**Audience:**

Yes

**Claims And Evidence:**

Yes

**Requested Changes:**

- please provide why high-resolution motion capture and wearable sensor data are available  for typical clinical or personal use settings
- please clarify why simpler architectures will not achieve similar or sufficient accuracy for practical purposes, given that the proposed architecture is so complex

**Strengths And Weaknesses:**

Strengths：
- The introduction of “multitask prospective rehearsal” is a key innovation, enabling real-time gait prediction without the need for retraining on new data.
- Real-world deployment with an ankle exoskeleton validates the model’s practical impact, suggesting it could significantly improve quality of life for individuals with mobility impairments.
- The paper includes ablation studies, stability analysis, and comparisons with various learning strategies, providing a strong validation of the model’s capabilities.

Weakness:
- I found that despite claims of practical applicability, the model has not been subjected to comprehensive clinical testing with real-world patients outside of lab conditions. This limitation calls into question the reliability and generalizability of the results. Without evidence from prolonged use in uncontrolled environments (such as daily life scenarios), it is unclear if the model could genuinely improve quality of life or withstand the demands of real-world wear and tear.
- The model relies on high-resolution motion capture and wearable sensor data, which are generally unavailable or impractical for typical clinical or personal use settings. The requirement for continuous, high-quality data acquisition adds significant complexity and cost, potentially limiting its practical deployment. Additionally, the model’s robustness to data artifacts, sensor misalignment, or low-quality sensor inputs is inadequately addressed, a notable oversight given real-world variability in sensor data.
- Despite claims that prospective rehearsal mitigates forgetting, the paper offers insufficient experimental evidence on how well the model truly retains knowledge over prolonged periods with changing tasks. Catastrophic forgetting remains a well-documented challenge in multitask learning, and without rigorous, long-term evaluations, the risk of task-specific knowledge erosion remains unaddressed.

---

> ### Author Response · Authors · 2024-11-03
> **Rebuttal: part 1**
>
> We sincerely appreciate your helpful review and valuable suggestions on our manuscript. Below, we address your comments point-by-point.
>
> **Clinical testing with patients.** We thank the reviewer for the comment. We agree with the reviewer that comprehensive clinical trials with real-world patients are important for establishing the reliability and generalizability of the proposed method. However, the results from our preliminary real-time hardware experiments, which tested the model across various motion conditions (Fig. 5 in the paper), indicate a promising foundation. These tests provided valuable insights into the model’s performance and robustness, which will guide our next steps. Of course, we plan to extend these trials to more subjects including patients, pending the ethics approval and will be a part of the future study. We are more than happy to discuss this limitation and plans for future work in the manuscript.
>
> **High-resolution motion capture and wearable sensor data:** We thank the reviewer for this comment. While we agree that obtaining optical motion capture data is impractical outside laboratory settings, wearable sensors such as IMUs offer a less costly and viable alternative to obtaining high-quality data. For practical purposes, such sensors can be embedded into the assistive device, making it a part of the assistive setup a patient is already using. As the reviewer pointed out, data artifacts and low-quality inputs can still pose a challenge. As a first step in addressing this, we demonstrated our model’s robustness to various levels of Gaussian noise added to the inputs, representing a common assumption for sensor noise (**Fig. 4C** in the paper). Additionally, we tested the model’s resilience under varying strengths of adversarial perturbations (**Fig. 4A** in the paper). Furthermore, we conducted real-time user experiments using wearable sensors in various motion conditions, where an ankle exoskeleton was controlled by joint profiles predicted by our model (**Fig. 5** in the paper). These tests offer promising insights into the model's performance and robustness in realistic settings. We would be glad to further emphasize these points in the manuscript to make it more accessible.

---

> > ### Author Response · Authors · 2024-11-03
> > **Rebuttal: part 2**
> >
> > **Catastrophic forgetting:** Thank you for the suggestion. In line with this suggestion and those from Reviewer Eak7, we have run additional experiments to illustrate how the rehearsal and task-specific prediction head architecture helps in dealing with catastrophic forgetting. Through ablation studies, we found that both the rehearsal and architectural choice are necessary for preventing the forgetting problem. Below, we showcase the results.
> >
> > - *RMSE* values of a model with task-specific prediction heads trained *without replay* on different tasks after it is trained for a particular task (Tab. 1).
> >
> > |               |                |           |             |   Train task  |              |                |
> > |---------------|----------------|:---------:|:-----------:|:-------------:|:------------:|:--------------:|
> > |               |                |  **Walk** | **Ramp up** | **Ramp down** | **Stair up** | **Stair down** |
> > |               	| **Walk**       | **0.054** |     0.14    |      0.16     |     0.177    |      0.18      |
> > |               	| **Ramp up**    |   0.454   |  **0.055**  |     0.147     |     0.134    |      0.176     |
> > | **Test task** 	| **Ramp down**  |   0.517   |    0.517    |   **0.055**   |     0.165    |      0.156     |
> > |               	| **Stair up**   	|   0.516   |    0.516    |     0.516     |   **0.054**  |      0.193     |
> > |               	| **Stair down** |   0.506   |    0.506    |     0.502     |     0.505    |    **0.066**   |
> >
> > - *RMSE* values of a model with *task-shared* prediction heads trained *with replay* on different tasks after it is trained for a particular task (Tab. 2).
> >
> > |               |                |           |             |   Train task  |              |                |
> > |---------------|----------------|:---------:|:-----------:|:-------------:|:------------:|:--------------:|
> > |               |                |  **Walk** | **Ramp up** | **Ramp down** | **Stair up** | **Stair down** |
> > |               | **Walk**       | **0.054** |  **0.062**  |   **0.067**   |   **0.071**  |    **0.078**   |
> > |               | **Ramp up**    |    0.15   |   **0.06**  |   **0.067**   |   **0.074**  |    **0.075**   |
> > | **Test task** | **Ramp down**  |   0.175   |     0.19    |   **0.064**   |   **0.083**  |    **0.083**   |
> > |               | **Stair up**   |   0.213   |    0.184    |     0.215     |   **0.071**  |    **0.083**   |
> > |               | **Stair down** |   0.235   |    0.231    |     0.204     |     0.228    |    **0.086**   |
> >
> > - *RMSE* values of a model with *task-specific* prediction heads trained *with replay* on different tasks after it is trained for a particular task (Tab. 3).
> >
> > |  |  |  | |   Train task  | | |
> > |---------------|----------------|:---------:|:-----------:|:-------------:|:------------:|:--------------:|
> > |               |                | **Walk**  | **Ramp up** | **Ramp down** | **Stair up** | **Stair down** |
> > |               | **Walk**       | **0.053** | **0.057**   | **0.058**     | **0.059**    | **0.062**      |
> > |               | **Ramp up**    | 0.454     | **0.057**   | **0.055**     | **0.056**    | **0.056**      |
> > | **Test task** | **Ramp down**  | 0.502     | 0.501       | **0.051**     | **0.056**    | **0.057**      |
> > |               | **Stair up**   | 0.553     | 0.551       | 0.551         | **0.053**    | **0.054**      |
> > |               | **Stair down** | 0.53      | 0.53        | 0.53          | 0.53         | **0.066**      |
> >
> >
> > A model trained without replay (Tab. 1) excels only on its current task, but suffers from catastrophic forgetting on previous tasks. In contrast, models with replay (Tab. 2 and 3) reduce forgetting. Furthermore, using task-specific prediction heads outperforms a shared head, further mitigating forgetting. Thus, both replay and task-specific heads help address forgetting.
> >
> > In addition, to systematically analyze the model performance as it is trained for multiple tasks, we also computed two metrics (similar to [3]) to measure how well the model performs on previously trained tasks after training it with new tasks.
> >
> > 1) The backward transfer is computed as
> > $$
> >     \mathit{BWT(t)} = \frac{1}{t-1}\sum_{j=1}^{t-1}\varepsilon_j(j) - \varepsilon_t(j)
> > $$
> > where $t$ is the latest task the model is trained for,
> > $\varepsilon_i(j)$ is the prediction error on task $j$ once the model was trained with the task $i$. $\mathit{BWT(t)}$ computes the change in performance of the model on previously trained tasks $j=\{1..t-1\}$ after it is trained for a new task $t =\{2...T\}$. Larger values of backward transfer are preferred.
> >
> > 2) The forgetting ratio for a task $t$ is computed as
> > $$
> >     \mathit{FR(t)} = \frac{\varepsilon_T(t)-\varepsilon_t(t)}{\varepsilon_t(t)}
> > $$
> >
> > where $T$ represents the final task on which the model was trained. $FR(t)$ gives the relative change in performance of the model on task, $t = \{1...T-1\}$, after it was trained with all available tasks. Smaller values of forgetting ratio are preferred.

---

> ### Author Response · Authors · 2024-11-03
> **Rebuttal: part 3**
>
> Following tables show the backward transfer and forgetting ratios of models trained without replay, and the ones trained with replay with different architectures.
>
> - Backward knowledge transfer, $BWT(t)$, (transfer of knowledge to previously learned tasks) when the model is trained with a new task, t (Tab. 4)
>
> | **Train task, t**             | **Walk**    | **Ramp up**     | **Ramp down**   | **Stair up**    | **Stair down**  |
> |-------------------------------|-------------|-----------------|-----------------|-----------------|-----------------|
> | **no replay**                 | --          | -0.043          | -0.065          | -0.079          | -0.098          |
> | **task-shared with replay**   | --          | -0.004          | -0.006          | -0.013          | -0.014          |
> | **task-specific with replay** | -- | **-0.002** | **-0.001** | **-0.002** | **-0.003** |
>
> - Forgetting ratios, $FR(t)$, of the model on a task, t after training it with all available tasks (Tab. 5)
>
> | **Evaluation task, t**        | **Walk**      | **Ramp up** | **Ramp down** | **Stair up**   | **Stair down** |
> |-------------------------------|---------------|-------------|---------------|----------------|------------------------------------|
> | **no replay**                 | 2.33          | 2.29        | 1.99          | 2.64           | --                                 |
> | **task-shared with replay**   | 0.42          | 0.25        | 0.27          | 0.16           | --                                 |
> | **task-specific with replay** | **0.16** | **0**  | **0.1**  | **-0.02** | --                                 |
>
> As can be seen, models with task-specific prediction heads and replay perform best in the face of continual task learning and dealing with catastrophic forgetting.
>
> **Simpler architectures:** Thank you for this comment. We conducted a comprehensive benchmarking of our approach with fully shared weights against a variety of models, including traditional linear models, Linear Support Vector Regression (SVR), Nonlinear SVR with a Radial Basis Function (RBF) kernel, Random Forest (RF), Decision Trees (DT), and neural network-based models such as Feedforward Artificial Neural Networks (FFAN) and Recurrent Neural Networks (RNNs) with Long Short-Term Memory (LSTM) and Gated Recurrent Unit (GRU) layers. Our results show that our method consistently outperforms these simpler models across all datasets.
>
>
> |                   |        LR    |     SVR (linear) |     Decision  tree |     Random  forest |     SVR  (RBF) |       kNN   |      FFANN  |       GRU   |       LSTM  |     ours (task-shared) |
> |-------------------|:------------:|:----------------:|:------------------:|:------------------:|:--------------:|:-----------:|:-----------:|:-----------:|:-----------:|:----------------------:|
> |     ENABL3        |       0.46   |         0.42     |          0.67      |           0.7      |        0.78    |       0.79  |       0.78  |       0.71  |       0.72  |            0.91        |
> |     Embry et. al. |       0.52   |         0.52     |          0.22      |          0.29      |         0.6    |       0.37  |       0.59  |       0.74  |       0.61  |            0.92        |
> |     Amputees      |     -0.11    |        0.24      |         0.4        |         0.45       |       0.45     |     0.49    |     0.21    |     -0.1    |     0.02    |           0.85         |

---

### Decision · Action_Editor_bqM9 · 2024-12-23

**Recommendation:** Accept as is

**Comment:**

The paper presents a multitask, continually adaptive model for human gait prediction in bionic prosthesis control, introducing the novel concept of multitask prospective rehearsal. While the reviewers raised concerns about limited novelty, practical validation, and competitiveness on public datasets, the authors addressed all major technical issues and provided additional experimental evidence to support their claims. The work is technically sound, aligns with TMLR’s standards for correctness over innovation, and addresses a relevant problem in a niche area. Given the reviewers’ consensus that the paper is of interest to the community and technically valid, I recommend acceptance.

**Audience:**

The paper addresses an important and relevant problem in the field of human behavior modeling for bionic prosthesis control, which is of interest to both the machine learning and healthcare communities. The proposed methodology, multitask prospective rehearsal, and its application to gait prediction in prosthetics could attract researchers working on continual learning, multitask learning, and adaptive models, as well as those involved in real-world applications such as healthcare and assistive technologies. While the focus is on a niche area, the technical contributions and innovative approach are relevant to a segment of TMLR's readership.

**Claims And Evidence:**

The authors provide extensive experimental results that demonstrate the effectiveness of their proposed framework, particularly in dealing with distribution shifts, noise, and adversarial attacks. The concerns raised by the reviewers, such as the potential for catastrophic forgetting and the complexity of the model, were addressed with additional experimental evidence. However, some reviewers pointed out that the practical validation of the framework, particularly regarding its performance on real-world tasks like imbalance and fall risks, was not fully explored. Despite this, the technical evidence presented is sound and supports the primary claims of the paper.